# Enhancing Decision-Making of Large Language Models via Actor-Critic

## Abstract

Large Language Models (LLMs) have achieved significant advancements in natural language processing tasks, yet they encounter challenges in complex decision-making scenarios that require long-term reasoning and alignment with high-level objectives. This paper introduces a novel gradient-free LLM-based Actor-Critic framework, termed `LAC`, which addresses these limitations by integrating both action generation and action evaluation mechanisms. Our approach employs two distinct critics: a language-based critic that provides context-sensitive feedback and a value-based critic that offers quantitative assessments of expected long-term rewards. This dual-critic architecture enhances decision-making by leveraging the complementary strengths of both critics, enabling contextually appropriate and more robust action selection. Additionally, we propose a gradient-free policy improvement method that reduces computational overhead, facilitating efficient updates to the actor's policy without the complexities of gradient backpropagation. We validate the effectiveness of `LAC` across diverse environments that cover both high-level action space (ALFWorld) and low-level action space (BabyAI-Text), demonstrating its superior performance compared to existing state-of-the-art methods. Our method outperforms other state-of-the-art baselines using the same 7B/8B open-source LLMs and even exceeds a strong baseline `ReAct` using GPT-4 in most settings. Our findings highlight the efficacy and generality of the dual-critic Actor-Critic framework in enhancing LLM-based decision-making.

## 1 Introduction

Large Language Models (LLMs) (Touvron et al., 2023; Jiang et al., 2023; Team et al., 2024) have demonstrated remarkable capabilities across a wide range of tasks in natural language processing, from text generation to question answering and summarization. Despite their strengths, LLMs often struggle in more complex decision-making tasks that require not only generating immediate action but also reasoning over long horizons and aligning actions with high-level objectives (Ahn et al., 2022; Yao et al., 2022b; Hao et al., 2023; Liu et al., 2023; Huang et al., 2022b). This raises a fundamental question: how can we efficiently leverage the rich prior knowledge encoded in LLMs to enable more reliable and effective sequential decision-making in diverse and complex environments?

Recent studies have explored various methods to improve LLM-based decision-making. Through the lens of reinforcement learning (RL) (Barto et al., 1989), these methods typically adopt either an *actor-only* or *critic-only* paradigm. In *actor-only* approaches, the LLM serves as an actor, generating actions based on its autoregressive next-token prediction capabilities (Ahn et al., 2022; Yao et al., 2022b; Huang et al., 2022a; Shinn et al., 2024). While such methods are simple and effective for short-term action generation, they often suffer from a lack of long-term planning. As a result, decisions may appear locally optimal but fail to achieve the overall task objective in more complex, multistep environments.

On the other hand, *critic-only* approaches use LLMs as critics to evaluate candidate actions based on predicted future trajectories (Hao et al., 2023; Liu et al., 2023; Fu et al., 2024; Brooks et al., 2024) and select the action with the best outcome. Although this allows for additional evaluation of actions, it may lead to suboptimal action selection when the sampled candidate actions do not include the optimal action and the predicted future trajectories deviate from reality. Furthermore, such methods often prioritize numerical assessments of actions, ignoring important contextual and

Figure 1: An illustrative explanation of our method LAC in ALFWorld. The histogram on the right shows the action probabilities of different methods. While actor ($\pi_{LLM}$) and critics (*lang-critic* $\mathcal{C}_{LLM}$, *value-critic* $\mathcal{Q}_{LLM}$) make mistakes at different time steps, LAC (ours) can select the correct action by integrating actor and critics. The LAC Inference step is detailed in Figure 2.

qualitative language-based information embedded in the task instructions. As shown in Figure 1, these two paradigms fall short of delivering optimal performance, as they fail to balance immediate action generation with long-term action evaluation. While previous works have attempted simple combinations of these roles (Zhang et al., 2023a), they lack a systematic analysis of the interaction between actor and critic, which is crucial for effective decision-making.

To address these limitations, we propose a novel gradient-free LLM-based Actor-Critic (LAC) framework. Unlike previous methods, our approach seamlessly integrates both *action generation* (actor) and *action evaluation* (critic) to significantly enhance decision-making capabilities. In LAC, the actor improves its policy by using two distinct critics: *lang-critic* and *value-critic*, which provide complementary types of feedback. The *lang-critic* leverages the LLM's natural language understanding capabilities to offer rich, interpretable, and context-sensitive feedback, ensuring that actions align with the task's high-level goals and that previous mistakes are avoided. The *value-critic*, on the other hand, provides numerical evaluations, estimating the long-term reward and ensuring that action distributions are quantitatively optimized. To obtain accurate numerical assessments of actions, we designed a novel *value-critic* estimation method that extracts the internal belief of LLMs on the optimality of candidate actions.

For intuition of the dual critics, consider a task where an agent must navigate through a virtual room to pick up an apple:

- The *lang-critic* might guide the agent with insights like "the apple is likely in the refrigerator in the kitchen", helping it prioritize actions that are contextually relevant (*e.g.*, move toward the refrigerator instead of wandering aimlessly).

- The *value-critic* evaluates the actions by estimating the long-term reward: "picking up the apple from the refrigerator has a 90% chance of completing the task", providing a precise quantitative judgment.

By integrating both critics, `LAC` selects actions that are both contextually appropriate (based on the *lang-critic*) and highly likely to succeed (based on the *value-critic*). As illustrated in Figure 1, our method outperforms previous *actor-only* and *critic-only* approaches, which often select suboptimal actions at different time steps. The integration of actor and dual critics in `LAC` leads to more effective decision-making and consistent task completion.

In addition, `LAC` introduces a novel *gradient-free policy improvement mechanism* that allows the actor to update its policy in the direction suggested by the critics without the computational burden of backpropagation. This not only enhances scalability but also makes the system practical for large-scale LLMs, enabling efficient decision-making in complex environments.

In summary, this work advances the state of LLM-based decision-making through the following key contributions:

- We introduce a dual-critic actor-critic framework, where a language-based critic provides contextual feedback and a value-based critic ensures quantitative optimization. This combination enables more robust decision-making by leveraging the complementary strengths of both critics.

- We design an effective value-based critic estimation approach that extracts internal information from LLMs to provide robust numerical evaluation for candidate actions.

- We propose a novel gradient-free policy improvement method that reduces computational overhead while effectively refining the actor's policy based on both qualitative and quantitative feedback from the dual critics.

- We demonstrate the effectiveness and generality of `LAC` across diverse environments, including high-level decision-making tasks in (ALFWorld, (Shridhar et al., 2021)) and low-level action space (BabyAI-Text, (Carta et al., 2023b)). Empirical results show that our approach consistently outperforms state-of-the-art methods such as `ReAct` Yao et al. (2022b) with GPT-4 (Achiam et al., 2023), even when using much smaller 7B/8B LLMs.

## 2 RELATED WORK

**Large Language Models for Sequential Decision-Making** Sequential decision-making is a fundamental ability of intelligent agents, involving generating a series of actions to achieve the goal (Barto et al., 1989; Littman, 1996; McCarthy et al., 1963; Bylander, 1994). Recently, LLM-based agents have been widely used for decision-making in many areas, which only needs some instructions or few-shot examples to generalize to completely new tasks (Huang et al., 2022b; Singh et al., 2023; Ding et al., 2023), thanks to the pre-training on large-scale dataset. According to the functionality, the LLMs that most previous work used mainly belong to two roles: actors, which take trajectories as input and output actions, and critics, which take both trajectories and actions as input and output evaluations of actions. Based on this classification, most of the earlier work in this line of research is *actor-only* (Ahn et al., 2022; Huang et al., 2022b; Yao et al., 2022b; Huang et al., 2022a; Shinn et al., 2024), *i.e.*, directly using the action generated by LLMs based on previous trajectory. Due to the auto-regressive nature of LLM, it does not do reasoning and planning explicitly. Accordingly, LLM with actor-only methods often struggles with complex tasks that require multiple steps of planning and reasoning (Huang & Chang, 2022; Mialon et al., 2023). To overcome this hurdle, another line of work, *critic-only* uses another LLM to evaluate each action by simulating the consequence of it and then choose the action with the best-simulated outcome (Hao et al., 2023; Liu et al., 2023; Fu et al., 2024). However, both *actor-only* and *critic-only* methods ignore the interrelation between actor and critic, prioritize one over the other, and insufficiently exploit the available knowledge from the actor and critic. Previous work that tries to combine actor and critic (Zhang et al., 2023b) only uses the language-based outputs of critics. Some work in other fields, such as decoding (Xie et al., 2024), also uses numerical outputs of critics, but it cannot be directly adapted to decision-making problems. To address these limitations, our method `LAC` integrates prior actor-only and critic-only methods and utilizes the merits of the actor-critic algorithm with the strengths of the LLMs.

**Large Language Models with Reinforcement Learning**  Classical sequential decision-making methods, such as Reinforcement Learning (RL), have been widely adopted in embodied environments (Schulman et al., 2017; Fujimoto et al., 2018; Huang et al., 2020; Dong et al., 2022). However, these RL-based methods are typically sample-inefficient and require lots of samples for training. On the other hand, LLMs that contain rich prior knowledge about the world may alleviate this burden. To combine RL and LLM, one straightforward way is to use LLMs as base models and add policy/value heads on top of LLMs (Carta et al., 2023a; Tan et al., 2024). Then use classical RL methods like PPO (Schulman et al., 2017) for training (Szot et al., 2023; Zhou et al.). However, these methods still require lots of training samples of the same tasks, which reduces the benefits of using LLM to some extent and contradicts our settings. There are also other paradigms for combining. RLEM (Zhang et al., 2024) adopts Q-learning (Watkins & Dayan, 1992) and an experience memory to update policies, but it may get stuck in the tasks with extremely sparse rewards like ALFWorld and BabyAI-Text. Retroformer (Yao et al., 2023) trains a smaller LLM with PPO to generate suitable prompts for a larger LLM for a specific task, while our method only needs a small model. ICPI (Brooks et al., 2024) uses LLMs to implement policy iteration by predicting future trajectories and accumulating future rewards, which may also struggle with sparse reward settings. We have compared it empirically in Section 5.

## 3 PRELIMINARY

In this section, we describe the task setting, previous *actor-only* methods, and *critic-only* methods to better understand prior work's limitations and the motivations for our LAC. For better understanding, we compare the frameworks of these methods in Figure 2.

**Task setup.** Consider a general setup of an agent interacting with an environment for achieving some given goals, *e.g.*, goal $g =$"put a clean egg in microwave" (from benchmark ALFWorld) or goal $g =$"pick up the green ball" (from benchmark BabyAI-Text). At time step $t$, the agent receives an observation $o_t \in \mathcal{O}$, which is described in natural language in this work, from the environment. The agent then takes an action $a_t \in \mathcal{A}$ that is sampled from some policy $\pi(a|g, h_t)$, where $h_t :=$ $(o_1, a_1, o_2, a_2 \cdots, o_t)$ is the history to the agent. During execution, there is no immediate reward. The environment will give a signal about whether the task was completed successfully or not only at the end of each episode. The agent has never seen the testing tasks before and can only try each task once in this work.

*Actor-only* **methods.** To solve the above tasks with large language models, one simple method is to directly use pre-trained LLMs as policy: $a_t \leftarrow \arg\max_a \pi_{LLM}(a|g, h_t)$, as shown in Figure 2 (a). We also provide detailed algorithm description of *actor-only* methods in Algorithm 2 of Appendix B.3, which can be implemented by simply injecting instructions or few-show examples to the prompt as shown in Yao et al. (2022b). Despite its simplicity, the actor-LLM $\pi_{LLM}$ generates actions solely relying on its auto-regression ability and it does not conduct long-term planning explicitly, which is typically necessary for sequential decision-making tasks. Additionally, this issue will be exacerbated when using smaller models like CodeLlama-7B (Roziere et al., 2023) and Mistral-7B (Jiang et al., 2023). This problem is verified in Section 5.

*Critic-only* **methods.** To handle the issue of lack of long-term planning in *actor-only* methods, another line of research resorts to *critic-only* methods (Hao et al., 2023; Liu et al., 2023; Fu et al., 2024). The basic idea of *critic-only* methods is to first sample several candidate actions from actor $\{a_t^1, a_t^2, \cdots, a_t^n\} \sim \pi_{LLM}(\cdot|g, h_t)$, then self-evaluate each candidate action by other LLMs and finally select the action with the highest evaluation value. We call it *critic-only* because only the critic's output is considered when choosing the final action. The self-evaluation procedure is the key to *critic-only* methods, which can adopt many approaches. For example, directly ask an LLM to evaluate the action candidate (Fu et al., 2024), or predict the future trajectory $u_t$ of each action candidate using an LLM as a forward model $f_{LLM}$ and use the future outcome as evaluation (shown in Figure 2 (b)), or use tree-search methods like Monte Carlo Tree Search (MCTS) (Kocsis & Szepesvári, 2006; Coulom, 2006) to expand each action candidate Hao et al. (2023). We also provide detailed algorithm description of *critic-only* methods in Algorithm 3 of Appendix B.4. Despite this progress, *critic-only* methods often neglect the knowledge of actor and the interaction between actor and critic, which may lead to ineffective decision-making.

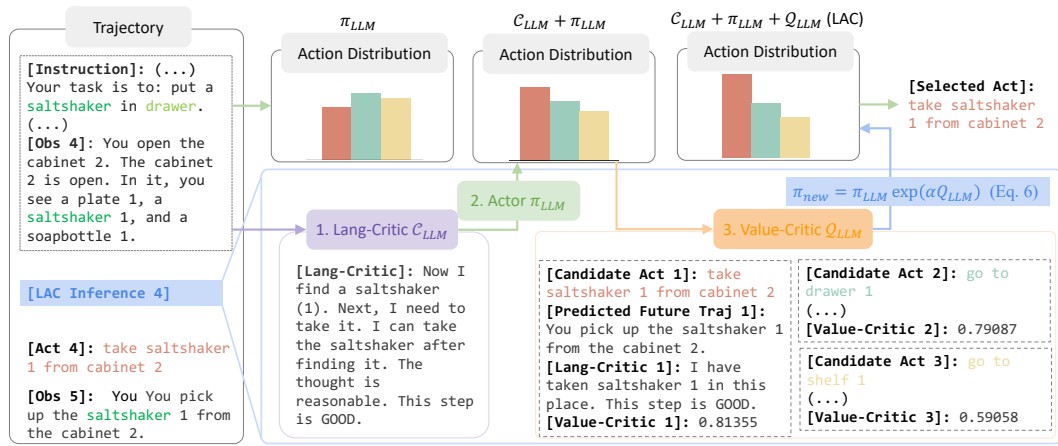

Figure 2: Framework of our LAC. At each time step, LAC selects an action via three steps: (1) given current goal and history, *lang-critic* $\mathcal{C}_{LLM}$ generates language-based judgments on previous actions; (2) the *actor* $\pi_{LLM}$ samples candidate actions based on the judgments; (3) the *value-critic* $\mathcal{Q}_{LLM}$ provides numerical evaluations for candidate actions by predicting future trajectories. Finally, the action distribution that integrates the actor and critics can be calculated in a gradient-free way.

## 4 METHOD

In this section, we present our LLM-based Actor-Critic (LAC) algorithm that integrates actor and critic to enhance the decision-making ability of large language models. The key idea behind LAC is to improve the *actor* $\pi_{LLM}$ based on the evaluations provided by two distinct but complementary critics: the *lang-critic* $\mathcal{C}_{LLM}$ and the *value-critic* $\mathcal{Q}_{LLM}$. The *lang-critic* $\mathcal{C}_{LLM}$ provides language-based evaluations that contain richer and more interpretable information for assessing actions, but they are difficult to quantify. The *value-critic* $\mathcal{Q}_{LLM}$ provides the numerical assessment in terms of long-term value or reward but may lack detailed explanations for assessment. By combining both natural language-based evaluations and numerical assessments, LAC ensures that actions are contextually relevant and optimized for long-term success. Figure 1 demonstrates how relying solely on one critic (either $\mathcal{C}_{LLM}$ or $\mathcal{Q}_{LLM}$) may still lead to suboptimal actions, whereas combining both critics avoids such mistakes.

The policy improvement process for $\pi_{LLM}$ involves two main steps: (1) improving $\pi_{LLM}$ with *lang-critic*'s language-based judgments over previous actions by injecting these judgments into $\pi_{LLM}$'s prompt so that $\pi_{LLM}$ could avoid previous mistakes and sample better candidate actions (Section 4.1); (2) further refining $\pi_{LLM}$ based on the *value-critic*'s numerical assessment through a gradient-free policy improvement procedure (Section 4.2). Our framework is shown in Figure 2. The overall algorithm is outlined in Algorithm 1. We also compare the frameworks of previous *actor-only* methods, *critic-only* methods, and our LAC, respectively, for better understanding in Figure 15 (a-c).

### 4.1 POLICY IMPROVEMENT WITH *lang-critic*

To enhance the actor's decision-making with contextually grounded feedback, we first prompt the LLM to generate judgments on previous action selections. Given the task goal $g$ and history $h_t$, the *lang-critic* generates a short evaluation sentence $c_t$ such as "I have found object-X. This step is GOOD" or "I should take object-X instead of object-Y first. This step is BAD." These judgments provide hints about whether and why the previous actions were appropriate, which helps guide the actor towards better candidate actions.

To improve the actor's policy $\pi_{LLM}$, we then condition $\pi_{LLM}$ on language-based evaluation $c_t$ before sampling candidate actions: $\{a_t^1, a_t^2, \cdots, a_t^n\} \sim \pi_{LLM}(\cdot|g, h_t, c_t)$. By incorporating $c_t$, the actor can generate better candidate actions through in-context learning, aligning more closely with task objectives and avoiding previous mistakes.

---

**Algorithm 1:** `LAC`: LLM-based Actor-Critic algorithm.

---

**Input:** current task goal $g$, history $h_t$, actor $\pi_{LLM}$, forward model $f_{LLM}$, language-based critic $\mathcal{C}_{LLM}$, value-based critic $\mathcal{Q}_{LLM}$, hyperparameter $\alpha$, candidate action size $n$.

**Output:** selected action $a_t^*$

1   $c_t \leftarrow \mathcal{C}_{LLM}(g, h_t)$;           ▷ generate language-based evaluations (Section 4.1)

2   $\{a_t^1, a_t^2, \cdots, a_t^n\} \sim \pi_{LLM}(\cdot|g, h_t, c_t)$;           ▷ generate candidate actions

3   **for** $i \leftarrow 1, 2, \cdots, n$ **do**

4      $u_t^i \leftarrow f_{LLM}(g, h_t, a_t^i)$;           ▷ imagine future trajectory

5      $\mathcal{Q}_{LLM}(g, h_t, a_t^i, u_t^i) \leftarrow \log \frac{P(y=+1|g,h_t,a_t^i,u_t^i)}{P(y=-1|g,h_t,a_t^i,u_t^i)}$;      ▷ calculate numerical evaluations

      (Section 4.2, Equation (3))

6   **end**

7   $\pi(a_t^i|g, h_t, c_t) \leftarrow \pi_{LLM}(a_t^i|g, h_t, c_t) \exp\left(\alpha \mathcal{Q}_{LLM}(g, h_t, a_t^i, u_t^i)\right)$;   ▷ update action distribution

    (Section 4.2, Equation (6))

8   $a_t^* \leftarrow \arg\max_{a_t^i} \pi(a_t^i|g, h_t, c_t)$

---

The key advantage of conditioning on $c_t$ is that it acts as an intermediate variable (Prystawski et al., 2024), helping the actor perform more effective reasoning. Similar to the Chain-of-Thought mechanism (Wei et al., 2022; Kojima et al., 2022), this enables the actor to adjust its policy based on the feedback provided by the *lang-critic*. For more examples of language-based evaluations, please refer to Table 14 and Table 15 of Appendix B.

### 4.2   POLICY IMPROVEMENT WITH *value-critic*

Next, we refine the actor's policy using the *value-critic* $\mathcal{Q}_{LLM}$, which provides numerical evaluations of each candidate action $a_t^i$. While the *lang-critic* focuses on providing contextually grounded feedback, the *value-critic* quantitatively estimates the probability of successfully completing the task after executing each action $a_t^i$. This value-based assessment is crucial for guiding the actor's decisions to align with long-term rewards, especially in tasks where immediate outcomes do not fully capture the consequences of an action.

In the following, we will first connect the value-critic to the agent's success probability of completing the task, then show how LLMs can be used to estimate this value-based evaluation, and finally derive a gradient-free policy improvement mechanism using the estimated value-based evaluation.

#### 4.2.1   CONNECT *value-critic* TO AGENT'S SUCCESS PROBABILITY

Let $\mathcal{Q}_{LLM}(g, h_t, a_t^i)$ be the value-based evaluation of each candidate action $a_t^i$ given the task goal $g$ and history $h_t$. Ideally, $\mathcal{Q}_{LLM}(g, h_t, a_t^i)$ should represent the cumulative rewards an agent can acquire after executing $a_t^i$, analogous to the action-value function in conventional RL algorithms. However, in the benchmarks we consider, only binary success or failure signals are provided at the end of each episode, with no intermediate rewards.

To model $\mathcal{Q}_{LLM}(g, h_t, a_t^i)$ similarly to action-value in RL, and to make it easy to estimate using LLMs, we employ a logistic function (Jordan et al., 1995). Let $P(y = +1|g, h_t, a_t^i) \in [0, 1]$ denote the probability of successfully completing the task goal $g$ after executing action $a_t^i$, where $y = +1$ represents a success signal at the end of the episode. Similarly, let $P(y = -1|g, h_t, a_t^i)$ represent the failure probability. We use the following logistic function to relate $P(y = +1|g, h_t, a_t^i)$ to $\mathcal{Q}_{LLM}(g, h_t, a_t^i)$:

$$P(y = +1|g, h_t, a_t^i) = \frac{1}{1 + \exp\left(-\mathcal{Q}_{LLM}(g, h_t, a_t^i)\right)}. \tag{1}$$

This formulation indicates that the value-based evaluation $\mathcal{Q}_{LLM}(g, h_t, a_t^i)$ is positively correlated with the success probability $P(y = +1|g, h_t, a_t^i)$. Higher $\mathcal{Q}_{LLM}(g, h_t, a_t^i)$ values map to a greater likelihood of success, allowing the critic to guide the actor's policy toward actions that maximize long-term success.

While other formulations could be used, we found that Equation (1) is both simple and effective for a wide range of tasks. For a comparison of alternative formulations, refer to Figure 8 in Appendix A.3.

### 4.2.2 ESTIMATE *value-critic* WITH LLMS

To estimate $\mathcal{Q}_{LLM}(g, h_t, a_t^i)$ using LLMs, we perform an equivalent transformation on Equation (1):

$$\mathcal{Q}_{LLM}(g, h_t, a_t^i) = \log \frac{P(y = +1|g, h_t, a_t^i)}{1 - P(y = +1|g, h_t, a_t^i)} = \log \frac{P(y = +1|g, h_t, a_t^i)}{P(y = -1|g, h_t, a_t^i)}. \tag{2}$$

With Equation (2), we can use the LLM to obtain value evaluation $\mathcal{Q}_{LLM}(g, h_t, a_t^i)$ via first estimating $P(y = \pm 1|g, h_t, a_t^i)$. The basic idea is to prompting the LLM to predict the success/failure probability given the current trajectory $(g, h_t)$ and action $a_t^i$. Specifically, we use the generated probabilities of special paired tokens that contain positive/negative meanings to indicate LLMs' belief in success/failure. For example, let the generated probability of "GOOD" or "SUCCESS" represent positive results $P(y = +1|g, h_t, a_t^i)$ and let the generated probability of "BAD" or "FAILURE" represent negative results $P(y = -1|g, h_t, a_t^i)$. The insight is that if the agent selects actions correctly, the LLM that are pre-trained via next-token prediction tends to increase the probability of *positive* tokens internally. Otherwise, the probability of *negative* tokens will increase. Finally, using Equation (2), we can calculate $\mathcal{Q}_{LLM}(g, h_t, a_t^i)$ for action $a_t^i$.

To improve the accuracy of $\mathcal{Q}_{LLM}(g, h_t, a_t^i)$, we introduce future trajectory rollouts using a forward model $f_{LLM}$, which can be implemented by prompting LLMs, *e.g.*, adding few-shot examples, or by fine-tuning on these examples. For each candidate action $a_t^i$, we roll out several future steps to predict the resulting trajectory $u_t^i = f_{LLM}(g, h_t, a_t^i)$. By considering the future trajectory $u_t^i$, we obtain more informed estimates of the success and failure probabilities, $P(y = \pm 1|g, h_t, a_t^i, u_t^i)$. This approach accounts for the delayed consequences of actions and ensures that $\mathcal{Q}_{LLM}(g, h_t, a_t^i)$ reflects the long-term value of each action:

$$\mathcal{Q}_{LLM}(g, h_t, a_t^i, u_t^i) = \log \frac{P(y = +1|g, h_t, a_t^i, u_t^i)}{P(y = -1|g, h_t, a_t^i, u_t^i)}. \tag{3}$$

Trajectory rollouts are especially important in tasks where the outcomes of actions may unfold over several steps. By simulating the future impact of actions, the value-critic provides a more accurate assessment, guiding the actor toward actions that maximize the probability of long-term success.

### 4.2.3 IMPROVE *actor* WITH *value-critic*

With the value-critic $\mathcal{Q}_{LLM}$, we can improve the actor's policy using the following optimization problem:

$$\max_\pi \mathbb{E}_{a_t^i \sim \pi(a_t^i|g, h_t, c_t)} \left[ \mathcal{Q}_{LLM}(g, h_t, a_t^i, u_t^i) \right] - \frac{1}{\alpha} \mathbb{D}_{KL} \left[ \pi(a_t^i|g, h_t, c_t) \| \pi_{LLM}(a_t^i|g, h_t, c_t) \right], \tag{4}$$

where $\alpha$ is a hyperparameter controlling the deviation from the original actor $\pi_{LLM}$. The KL-divergence term prevents the new actor $\pi$ from deviating too far from the original policy, balancing the actor's prior knowledge and the value-critic's guidance.

Following prior work (Rafailov et al., 2024; Go et al., 2023; Peng et al., 2019; Jain et al., 2013; Peters & Schaal, 2007), we can show that the optimal solution to the KL-constrained maximization objective in Equation (4) takes the following form:

$$\pi(a_t^i|g, h_t, c_t) = \frac{1}{Z(g, h_t, c_t)} \pi_{LLM}(a_t^i|g, h_t, c_t) \exp\left(\alpha \mathcal{Q}_{LLM}(g, h_t, a_t^i, u_t^i)\right), \tag{5}$$

where $Z(g, h_t, c_t) = \sum_{a_t^i} \pi_{LLM}(a_t^i|g, h_t, c_t) \exp\left(\alpha \mathcal{Q}_{LLM}(g, h_t, a_t^i, u_t^i)\right)$ is the partition function. Please refer to Appendix B.1 for a complete derivation. As the partition function does not depend on action $a_t^i$, we can ignore it in practice:

$$\pi(a_t^i|g, h_t, c_t) \propto \pi_{LLM}(a_t^i|g, h_t, c_t) \exp\left(\alpha \mathcal{Q}_{LLM}(g, h_t, a_t^i, u_t^i)\right). \tag{6}$$

We simply take the action with maximum proportion $a_t \leftarrow \arg\max_{a_t^i} \pi(a_t^i|g, h_t, c_t)$. It is worth mentioning that if we let $\alpha = 0$ and remove $c_t$ in Equation (6), we recover *actor-only* methods. And if we let $\alpha \to +\infty$, we recover *critic-only* methods that use our action-value estimation approach.

There are two key advantages of using Equation (6). Firstly, it updates the action distribution of *actor* $\pi_{LLM}$ in the direction suggested by *critic* $\mathcal{Q}_{LLM}$ in a gradient-free way, which achieves policy improvement with much lower computation burden compared to gradient-based methods, especially when the actor is realized by a large model. Secondly, the action distribution of the new actor $\pi$ is a balanced integration of the actor's prior based on past information and the critic's posterior based on predicted future information.

## 5 EXPERIMENTS

In this section, we benchmark our method `LAC` on benchmarks that cover both high-level action space (ALFWorld (Shridhar et al., 2021)) and low-level action space (BabyAI-Text (Chevalier-Boisvert et al., 2019)). We evaluate the effectiveness of `LAC` by answering the following questions: (1) Can `LAC` outperform other decision-making with LLMs methods? (Section 5.2) (2) How does each component of `LAC` contributes to its performance? (Section 5.3) (3) How do different large language models influence performance? (Section 5.2 and Section 5.3) (4) Is our method computationally consuming? (Section 5.4).

### 5.1 EXPERIMENT SETUP

We compare our method with various decision-making with LLMs baselines, which can be largely classified into *actor-only* and *critic-only* methods. For more details of baselines, please refer to Appendix C.2.

***Actor-only* methods.** ReAct (Yao et al., 2022b) combines reasoning and acting in the interaction with the environment and leverages the reasoning capabilities of LLMs to increase the probability of the LLM acting correctly as an actor.

***Critic-only* methods.** RAP(Hao et al., 2023) utilizes LLMs as actor and world models and adopts tree-search planning methods to evaluate each possible action candidate. ICPI (Brooks et al., 2024) implements policy iteration using LLMs by predicting future trajectories and selecting the action with the highest predicted cumulative rewards. RAFA (Liu et al., 2023) evaluates each action candidate by tree-search and selects the action that may complete the most sub-goals.

We evaluate `LAC` on two decision-making benchmarks with high-level actions and low-level actions, respectively.

**Benchmark with high-level actions: ALFWorld.** ALFWorld (Shridhar et al., 2021) is a widely used text-based household environment with 134 different tasks, which require the agent to achieve a goal through a sequence of high-level actions, *e.g.* "go to place-X", "take object-Y from place-X", *etc.*. The main challenge of this benchmark is to locate the target object and fulfill household work with commonsense knowledge of LLMs. Following `ReAct`, we evaluate all 134 unseen evaluation games in a task-specific setup.

**Benchmark with low-level actions: BabyAI-Text.** BabyAI-Text (Carta et al., 2023b) is a Grid World environment that extended from the BabyAI platform (Chevalier-Boisvert et al., 2019), in which the agent and objects are placed in a room of $8 \times 8$ tiles. The agent has 6 primitive actions: turn left, turn right, go forward, pick up, drop, toggle, to solve a task described in natural language (*e.g.* "Pick up the red box"). These tasks could be difficult because agents have to make a long-term plan, avoid obstacles and find a short path to target objects based on partial observations that are described in natural language

To show the stability of `LAC`, we adopt four open-source large language models from different organizations: CodeLlama-7B (Roziere et al., 2023), Mistral-7B (Jiang et al., 2023), Gemma-7B (Team et al., 2024), and Llama-3-8B (Meta, 2024a).

### 5.2 PERFORMANCE

We report the results of our method `LAC` compared with other baselines in ALFWorld and BabyAI-Text in Figure 3 and Figure 6 of Appendix A.1. For all experiments, we set the temperature of LLMs to 0, hence the generation is deterministic. For this reason, there is no error bar in the figure.

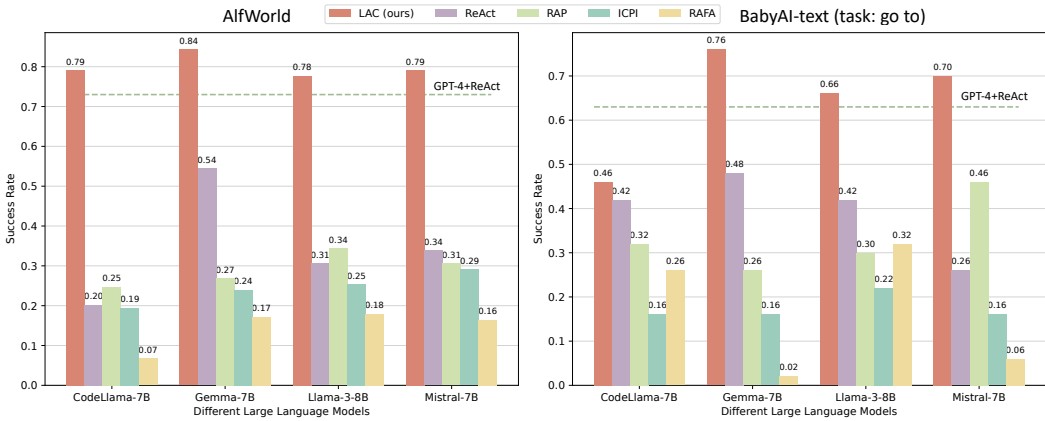

Figure 3: Performance of our LAC compared with various baselines in benchmarks ALFWorld and BabyAI-Text.

LAC outperforms all other baselines on both ALFWorld and BabyAI-Text across different LLMs, and LAC is even better than GPT-4+ReAct in most settings, which validates our method's effectiveness and stability.

LAC's superior performance stems from its balanced integration of the strengths of both actor and critic. While actor-only (*e.g.*, ReAct) methods excel in short-term actions, they often struggle with long-term reasoning. In contrast, critic-only (*e.g.*, RAP) methods conduct explicit reasoning but might mispredict future trajectories and lead to even worse action selection occasionally compared with actor-only methods. LAC addresses these limitations by balancing the actor's action generation and the critic's evaluation. We have provided illustrative examples for ALFWorld and BabyAI-Text in Figure 1 and Figure 13 respectively. In summary, actor-only and critic-only methods make mistakes at different time steps, our LAC can select the correct action.

Regarding the performance of LAC with different base models, we highlight two key findings: (1) Our method is general and can be adapted to various base models, and (2) stronger base models, such as Gemma-7B, demonstrate higher performance when integrated with our approach. However, due to the incomplete public availability of training details for these base models, further in-depth analysis will require additional investigation.

## 5.3 ABLATION STUDIES

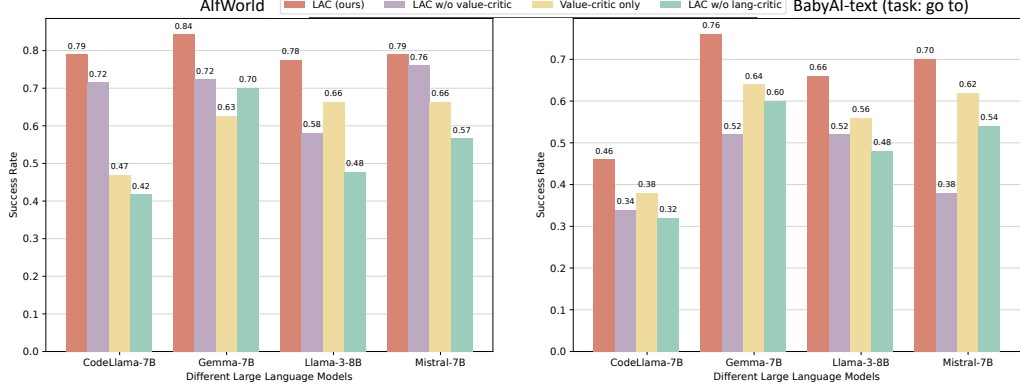

Figure 4: Ablation studies in benchmarks ALFWorld and BabyAI-Text.

To investigate the contributions of each component of LAC, we conduct elaborate ablation studies. There are two main components that characterize our method: (1) the integration of actor $\pi_{LLM}$ and

lang-critic $\mathcal{C}_{LLM}$ before action generation and (2) the integration of actor $\pi_{LLM}$ and value-critic $\mathcal{Q}_{LLM}$ after action generation. Therefore, to show the contribution of each component, we design the following ablation studies: (1) LAC *w/o* lang-critic removes the lang-critic $\mathcal{C}_{LLM}$ from LAC as well as the integration before action generation; (2) LAC *w/o* value-critic removes the value-critic $\mathcal{Q}_{LLM}$ from LAC as well as the integration after action generation; (3) Value-critic-only only uses value-critic $\mathcal{Q}_{LLM}$ for decision-making.

We report the result in Figure 4. LAC is better than all other variants in both ALFWorld and BabyAI-Text. Specifically, the performance decrease in LAC *w/o* lang-critic and LAC *w/o* value-critic compared with LAC verify the effectiveness of lang-critic $\mathcal{C}_{LLM}$ and value-critic $\mathcal{Q}_{LLM}$, respectively. And the result that Value-critic-only performs worse than LAC also suggests the necessity for integrating actor and critic.

### 5.4 COMPUTATIONAL COST ANALYSIS

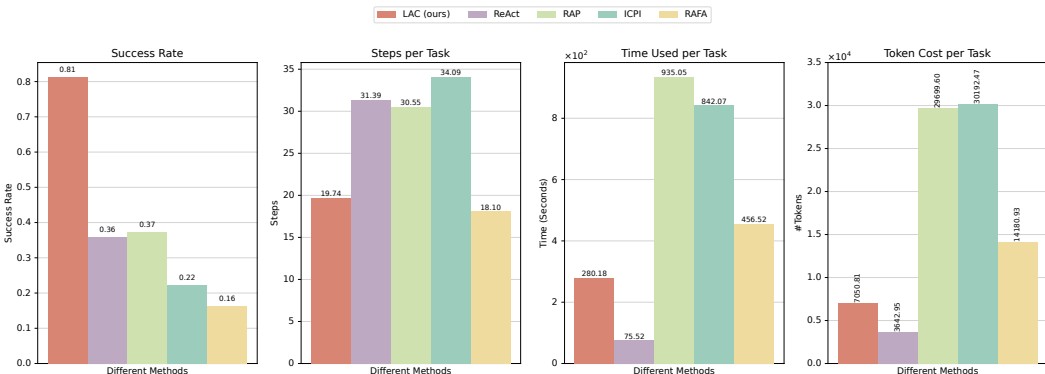

Figure 5: Computational cost analysis of LAC and baselines.

Our method integrates the actor and two critics, which may bring extra computational cost per step. In Figure 5, we compare computational costs concerning the number of tokens spent and running time between LAC and other baselines. Specifically, though LAC has a higher computational cost per step due to the extra inference procedure of critics and the forward model, the total cost of LAC is still lower than most LLM-based baselines because LAC requires fewer steps to finish each task.

## 6 DISCUSSION

In this work, we introduce a novel LLM-based Actor-Critic algorithm LAC that integrates the ability of actors and critics as well as exploits the strong prior knowledge in LLMs for sequential decision-making. Compared with previous *actor-only* and *critic-only* methods, LAC achieves high performance on ALFWorld and BabyAI-Text even using small open-source LLMs.

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

## A    EXTRA RESULTS

### A.1    RESULTS OF OTHER TASKS IN BABYAI-TEXT

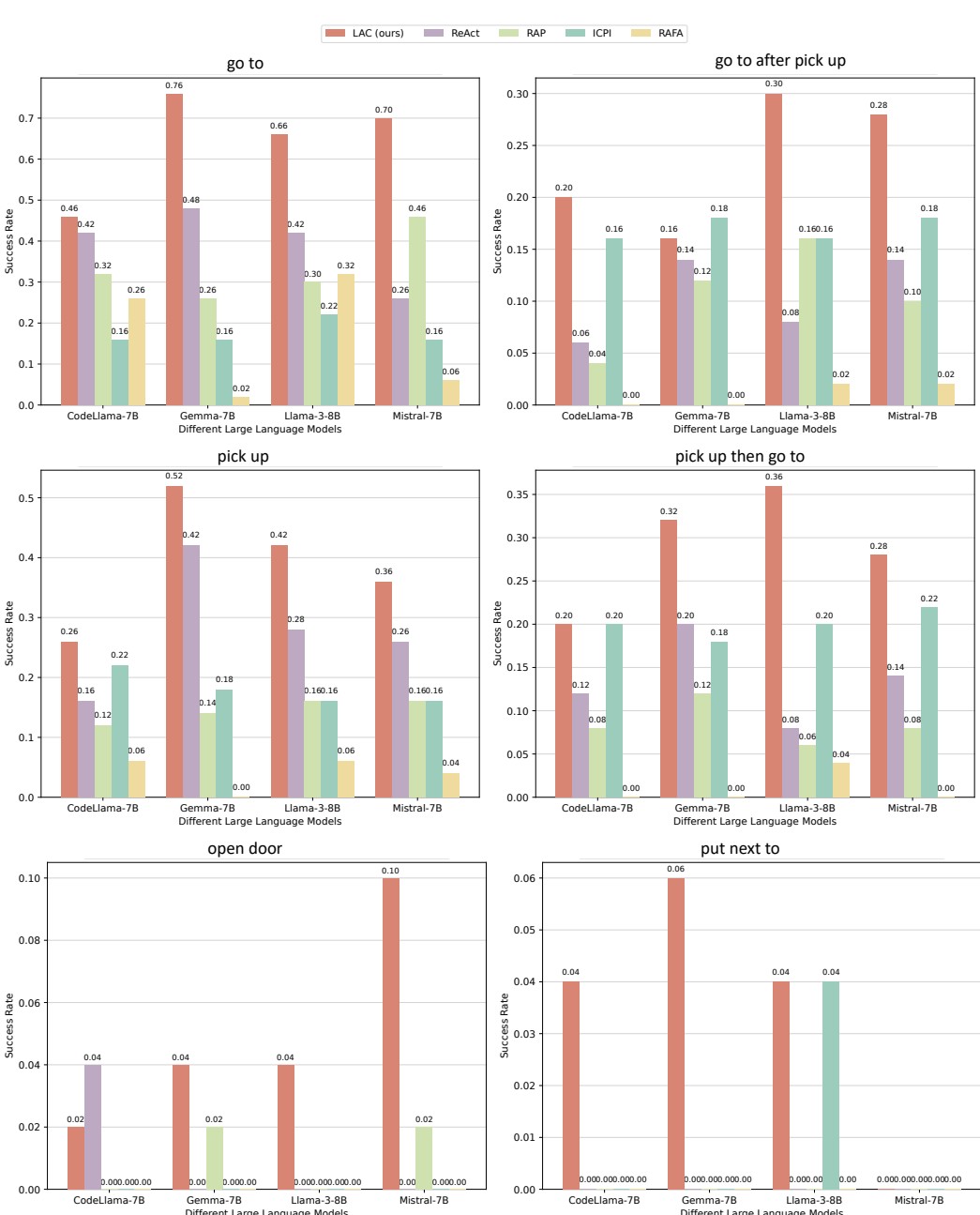

Figure 6: Performance of our LAC compared with various baselines in all tasks from BabyAI-Text.

For a complete comparison, we show the performance of LAC and baselines in other tasks from BabyAI-Text in Figure 6. Our LAC outperforms all other baselines, which further validates the effectiveness of LAC.

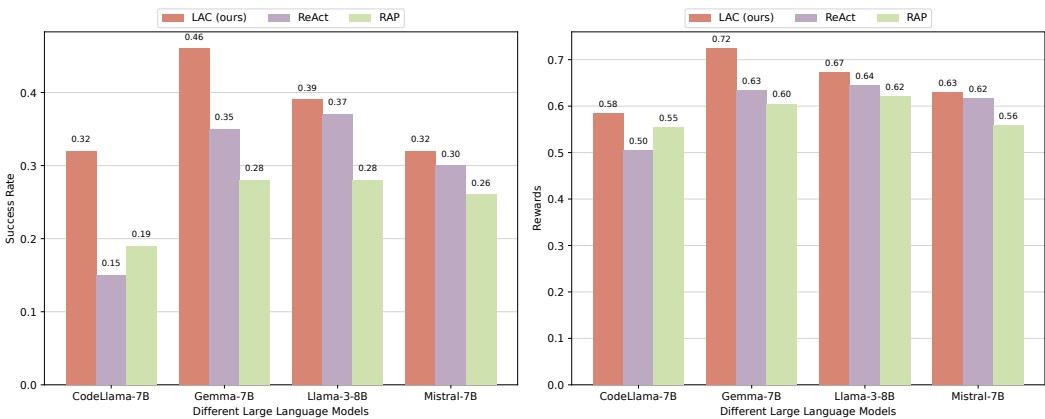

Figure 7: Performance of `LAC` in benchmark WebShop

## A.2 RESULTS OF `LAC` IN WEBSHOP

We have conducted new experiments using the WebShop benchmark (Yao et al., 2022a), which presents a scenario with a potentially infinite action space. This benchmark requires an agent to purchase a product based on specific instructions (e.g. "I need a long clip-in hair extension which is natural looking, and price lower than 20.00 dollars") through web interactions (e.g. search "long clip-in hair extension", choose buttons such as "click [item ID]" or "back to search"). Within this context, the "search" and "click" actions can indeed lead to an unbounded set of potential actions, as the agent can continuously refine its queries and selections based on dynamic web results.

The results of our experiments are detailed in Figure 7. We found that our method, `LAC`, consistently outperforms other baselines, in terms of both accumulated reward and success rate across various base models. This demonstrates the robustness of our method in handling more complex and open-ended action spaces.

## A.3 RESULTS OF USING OTHER DEFINITION OF $\mathcal{Q}_{LLM}$

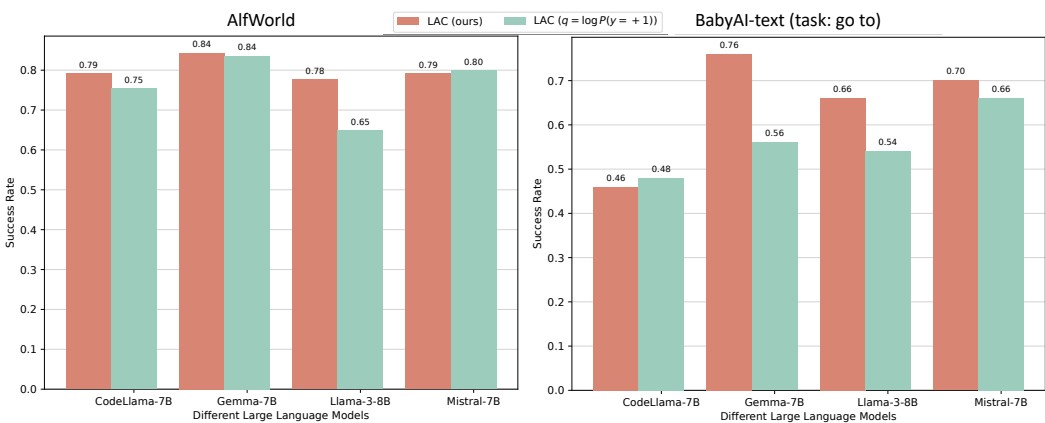

Figure 8: Performance of `LAC` when using different definition of value-critic $\mathcal{Q}_{LLM}$

In `LAC` we define lang-critic $\mathcal{Q}_{LLM}$ as $\mathcal{Q}_{LLM}(g, h_t, a_t^i) = \log \frac{P(y=+1|g,h_t,a_t^i)}{P(y=-1|g,h_t,a_t^i)}$. There are also other definitions, for example, the simplest variant is $\mathcal{Q}_{LLM}(g, h_t, a_t^i) = \log P(y = +1|g, h_t, a_t^i)$.

In this subsection, we provide a performance comparison between them in Figure 8. `LAC` outperforms the variant in most situations across tasks and models. We speculate that this is because `LAC` uses more information, *i.e.*, both $P(y = +1|g, h_t, a_t^i)$ and $P(y = -1|g, h_t, a_t^i)$, than the variant, and the

evaluation might be more accurate and more stable. There might be other definitions of $\mathcal{Q}_{LLM}$ and among them, our $\mathcal{Q}_{LLM}$ is simple and effective.

### A.4  COMPARISON OF LAC WITH MORE BASELINES ON ALFWORLD

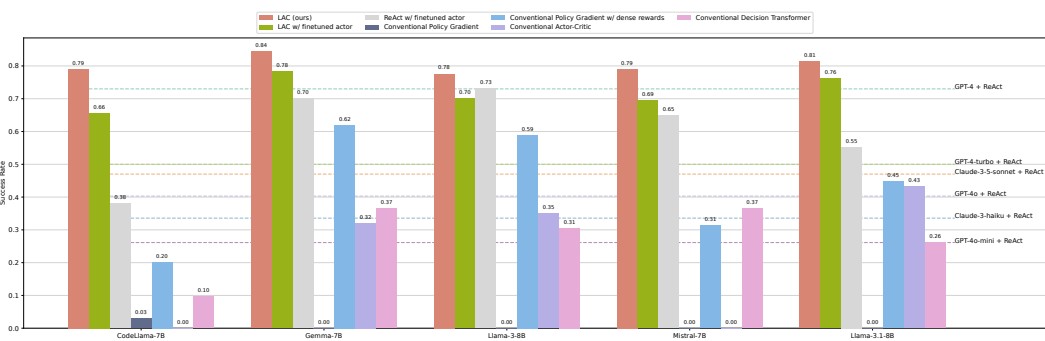

Figure 9: Performance of our LAC and LAC's variants compared with various baselines in benchmark ALFWorld.

In this subsection, we compare LAC with more baselines including some traditional RL methods implemented using LLMs on ALFWorld (Shridhar et al., 2021). The comparison is shown in Figure 9.

While in LAC we fine-tune the critic using a few trajectories, it is also possible to fine-tune the actor to generate actions using those trajectories. Therefore, a potential baseline could be fine-tuning the actor in *actor-only* method. To demonstrate the improvement brought by fine-tuning the actor, we fine-tune the actor in ReAct (Yao et al., 2022b) and show the results in Figure 9. We also show the results of LAC w/ fine-tuned actor in Figure 9. In brief, ReAct w/ fine-tuned actor is a strong baseline compared with other baselines, but still inferior to our method LAC and LAC w/ fine-tuned actor. Compared to LAC, the underperformance of LAC w/ fine-tuned actor arises from its tendency to overfit the training trajectories. This overfitting causes the actor to favor actions that are more frequent in the dataset, potentially leading to suboptimal action selection.

For example, in the ALFWorld training dataset, the action "take an apple from X" occurs frequently. After fine-tuning, the actor may disproportionately generate this action, even when it is irrelevant to the current goal. One case is that the current goal is to "heat some egg and put it in the garbage can". When the agent sees an "apple 2" in "fridge 1", it generates and selects an irrelevant action "take apple 2 from fridge 1", which does not align with the task.

This tendency towards overfitting arises because the complexity of the policy function, which maps states $s$ to actions $a$, often exceeds that of the critic. The policy often has to capture a wide variety of potential actions for each state, particularly in complex environments. However, the quite limited training dataset in our setting restricts its ability to generalize effectively, resulting in memorization of specific actions rather than flexible decision-making. In contrast, our critic, which includes a world model for rollout and an evaluation function, focuses on capturing more predictable dynamics of the environment and simpler evaluation criteria. This typically requires simpler mappings than those needed for the policy, thus avoiding overfitting.

We also include some LLM-based RL variants as baselines to show the superiority of LAC over conventional RL algorithms. We design three LLM-based RL variants that are built upon pre-trained LLMs and directly extract actions/values information from LLMs without adding action/value heads, namely Conventional Policy Gradient, Conventional Policy Gradient w/ dense rewards and Conventional Actor-Critic in Figure 9.

For the implementation of the Conventional Policy Gradient, we need the probability of actions and the returns. To obtain the probability of actions, we directly use LLM to compute the conditional probability of each token in action $a_i = [w_1, w_2, \cdots, w_{|a_t|}]$ given the goal $g$, history $h_t$ and then calculate their product:

$$\pi(a_t|g, h_t) = \Pi_{j=1}^{|a_t|} P_{LLM}(w_j|g, h_t, w_{<j})$$

in which $P_{LLM}(w_j|g, h_t, w_{<j})$ is the probability of token $w_j$ given goal $g$, history $h_t$ and previous tokens $w_{<j}$ computed by LLM. Then we regard the cumulative future rewards as the return $G_t$, which is $+1$ for successful trajectories and $-1$ for failed trajectories in the tasks we considered. Finally, the gradient of policy is $\mathbb{E}[\sum_t \nabla\log\pi(a_t|g, h_t)G_t]$. Conventional Policy Gradient w/ dense rewards is similar to Conventional Policy Gradient except that we manually add intermediate rewards for each step, and then use the cumulative future rewards as the return $G_t$.

For the implementation of the Conventional Actor-Critic, we additionally need a critic to estimate action values. As it is possible to train a new value head using only 18 trajectories, we instead approximate the action value similar to $\mathcal{Q}_{LLM}$ in our method LAC, *i.e.* $\mathcal{Q}_{LLM}(g, h_t, a_t, u_t) = \log\frac{P_{LLM}(y=+1|g,h_t,a_t,u_t)}{P_{LLM}(y=-1|g,h_t,a_t,u_t)}$, in which $P_{LLM}(y = \pm1|g, h_t, a_t, u_t)$ is the output probability of special positive/negative tokens like GOOD or BAD that indicate positive/negative results as LLM's belief on success/failure. Finally, the gradient of policy is $\mathbb{E}[\sum_t \nabla\log\pi(a_t|g, h_t)\mathcal{Q}_{LLM}(g, h_t, a_t, u_t)]$.

In summary, Conventional Policy Gradient exhibits almost all zero performance, which is due to the extremely sparse reward problems, compared with Conventional Policy Gradient w/ dense rewards. Conventional Actor-Critic demonstrates non-zero performance only on some stronger LLMs like Gemma-7B (Team et al., 2024), Llama-3-8B (Meta, 2024a) and Llama-3.1-8B (Meta, 2024b), which may be because the optimization method of conventional actor-critic is not suitable in insufficient data settings.

In addition to the aforementioned LLM-based RL variant, Decision Transformer (Chen et al., 2021) is also a potential solution in combining RL and transformer-based LLMs. We fine-tune pretrained LLMs in a similar way as conventional decision transformers. We construct a dataset using decision-transformers' trajectory representation: $\tau = [R_1, s_1, a_1, R_2, s_2, a_2, \cdots]$, in which $R_t$ is return-to-go, *i.e.*, $+1$ for successful trajectories and -1 for failed trajectories in our extremely sparse reward settings. Then we fine-tune LLMs with next-token prediction loss on these trajectories. During execution, we insert $+1$ before state $s_t$ to specify the desired outcome. The results are shown in Figure 9 as Conventional Decision Transformer. In short, Conventional Decision Transformer exhibits a similar performance to ReAct, which may be because the 18 trajectories are insufficient for fine-tuning decision transformers.

Our method LAC is better than all considered baselines because of its ability to handle extremely sparse reward problems using LLM's prior knowledge and to fully utilize insufficient data.

### A.5 MORE ABLATIONS OF LAC IN ALFWORLD AND BABYAI-TEXT

The *value-critic* relies on future trajectory predictions to improve the accuracy of its evaluations. By predicting future trajectories, the critic considers long-term consequences and evaluates actions more effectively, which ultimately leads to better decision-making.

For a full comparison, here we conducted an extra experiment for LAC *w/o* rollout, in which the *value-critic* generate value-based evaluations without future trajectory predictions. The results, included in Figure 10 and Figure 11, show that LAC *w/o* rollout consistently underperforms compared to the full LAC across various base models. This finding emphasizes the importance of future trajectory predictions for accurate evaluations.

### A.6 COMPUTATIONAL COST ANALYSIS OF LAC WITH MORE BASELINES IN ALFWORLD

In this subsection, we demonstrate the computation cost of LAC and other baselines in Figure 12. We show the success rate, steps per task, time used per task, and token cost per task respectively. Specifically, though LAC has a higher computational cost per step due to the extra inference procedure of critics and the forward model, the total cost is still lower than most LLM-based baselines because LAC has a higher success rate and requires fewer steps to finish each task.

### A.7 ILLUSTRATION OF BABYAI-TEXT

We should the illustrative example of BabyAI-Text in Figure 13.

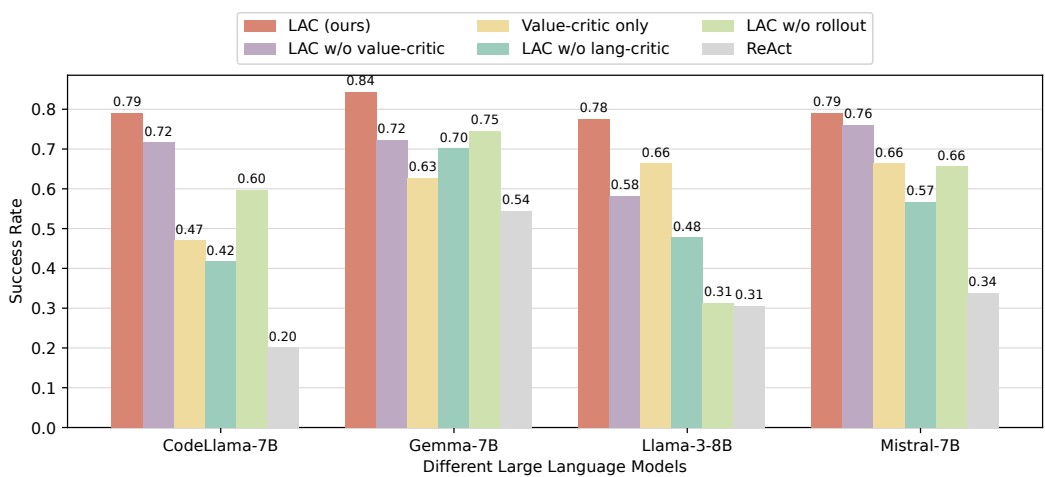

Figure 10: More ablation studies of `LAC` in ALFWorld

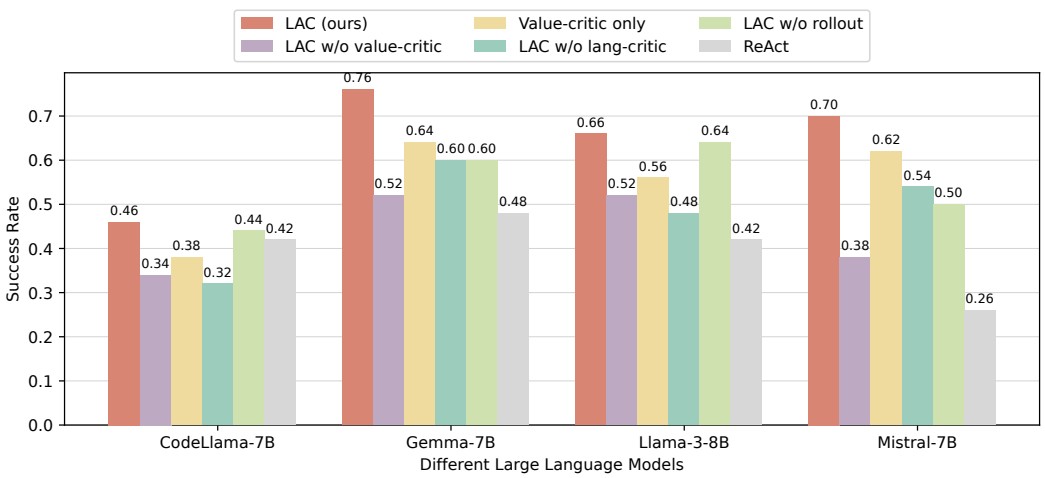

Figure 11: More ablation studies of `LAC` in BabyAI-Text

## A.8 RESULTS OF DIFFERENT CRITIC IMPROVEMENT METHODS

To improve the critics given only several examples, we can fine-tune the open-source models via next-token prediction. Please refer to Appendix B.2 for more fine-tuning details. To show the effectiveness of fine-tuning, we present the performance of `LAC` and other variants on task "go to" and "pick up" from BabyAI-Text when we just add these examples into the prompt, *i.e.*, in-context learning, in Table 1 and Table 2. We also show the the performance improvement if we do fine-tuning in the parentheses of Table 1. This result indicates that (1) fine-tuning can incorporate extra knowledge into LLMs better than in-context learning in our case (2) both of our two critics can benefit from fine-tuning. It is worth mentioning that our `LAC` still outperforms baselines without fine-tuning.

Table 1: Performance of two critic improvement methods: in-context learning or fine-tuning.

|  | CodeLlama-7B | Gemma-7B | Llama-3-8B | Mistral-7B |
|---|---|---|---|---|
| `LAC` | 0.30 (↑ 0.16) | 0.62 (↑ 0.14) | 0.32 (↑ 0.34) | 0.24 (↑ 0.46) |
| `LAC` *w/o* lang-critic | 0.30 (↑ 0.02) | 0.58 (↑ 0.02) | 0.38 (↑ 0.10) | 0.26 (↑ 0.28) |
| `LAC` *w/o* value-critic | 0.28 (↑ 0.06) | 0.48 (↑ 0.04) | 0.42 (↑ 0.10) | 0.10 (↑ 0.28) |
| Value-critic-only | 0.42 (↑ 0.04) | 0.40 (↑ 0.24) | 0.34 (↑ 0.22) | 0.38 (↑ 0.24) |

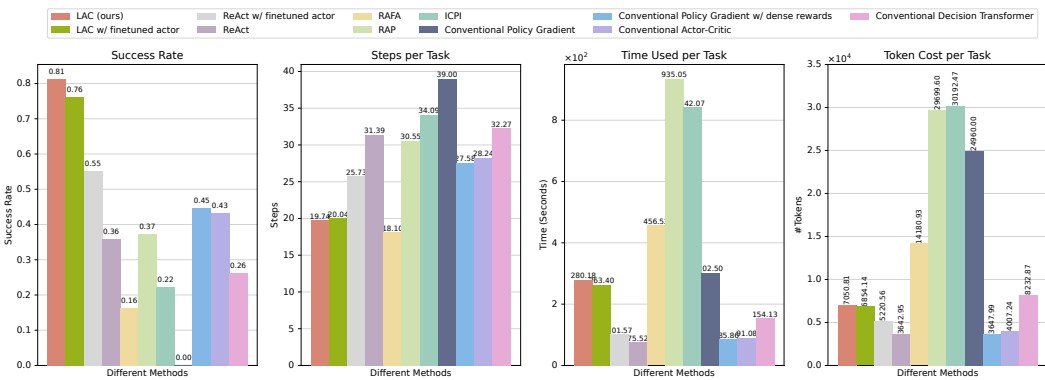

Figure 12: Computational cost analysis of our `LAC` compared with various baselines in benchmarks ALFWorld. Though LAC may have a higher computational cost per step due to the extra inference procedure of critics and the forward model, the total cost of LAC is still lower than most LLM-based baselines because LAC requires fewer steps to finish each task.

Table 2: Performance of two critic improvement methods: in-context learning or fine-tuning.

|  | CodeLlama-7B | Gemma-7B | Llama-3-8B | Mistral-7B |
|---|---|---|---|---|
| `LAC` | 0.20 ($\uparrow$ 0.06) | 0.22 ($\uparrow$ 0.20) | 0.34 ($\uparrow$ 0.08) | 0.20 ($\uparrow$ 0.16) |
| `LAC` *w/o* lang-critic | 0.16 ($\uparrow$ 0.08) | 0.32 ($\uparrow$ 0.04) | 0.32 ($\uparrow$ 0.04) | 0.22 ($\uparrow$ 0.06) |
| `LAC` *w/o* value-critic | 0.12 ($\uparrow$ 0.14) | 0.36 ($\uparrow$ 0.20) | 0.28 ($\uparrow$ 0.06) | 0.26 ($\uparrow$ 0.04) |
| Value-critic-only | 0.22 ($\uparrow$ 0.04) | 0.24 ($\uparrow$ 0.26) | 0.16 ($\uparrow$ 0.16) | 0.16 ($\uparrow$ 0.26) |

### A.9 ANALYSIS OF THE FINE-TUNING PROCESS IN `LAC`

In order to improve the quality of the language-based critic generated by LLM, we finetune the LLM that generates the critic. In this section, we analyze the finetuning process, showing the impact of finetuning on critic prediction, as well as the impact of different data amounts and positive and negative sample ratios on task success rates. The comparison can be seen in Figure 14.

In Figure 14 (a), we show the influence of fine-tuning data size. We use 9, 18, 27 and 36 trajectories to fine-tune LLMs, and show the final success rate on 134 evaluation tasks. In summary, larger data sizes (27 or 36 trajectories) generally bring higher success rate, while small data sizes (18 and even 9 trajectories in some cases) are already enough for `LAC` to achieve outperformance.

Figure 14 (b) shows the influence of different positive/negative sample ratio (positive:negative = 0:1, 1:3, 1:1, 3:1 and 1:0) on final performance. We keep the total number of samples the same and just change positive/negative ratio. In short, our `LAC` is robust to reasonable positive/negative ratios (*e.g.* 1:3, 1:1, 3:1), while `LAC` based on CodeLlama-7B (Roziere et al., 2023) and Gemma-7B (Team et al., 2024) even perform better when given all positive samples (1:0).

Figure 14 (c) shows the learning curves of the fine-tuning process. We plot the next prediction loss and positive/negative tokens prediction accuracy for CodeLlama-7B (Roziere et al., 2023). In short, as the next token prediction loss decreases during fine-tuning, the accuracy of predicting the special tokens (GOOD or BAD) increases, which exhibits the effect of the fine-tuning process.

### A.10 RESULTS OF DIFFERENT HYPER-PARAMETER $\alpha$

The hyper-parameter $\alpha$ in Equation (4) controls the deviation from the original actor $\pi_{LLM}$. In this subsection, we grid-search this hyper-parameter over $\{1/2, 1, 2, 5, 10\}$ in task "go to" of BabyAI-Text, then we fix $\alpha$ for other tasks: $\alpha = 1$ for model CodeLlama-7B, $\alpha = 2$ for model Gemma-7B, $\alpha = 2$ for model Llama-3-8B and $\alpha = 10$ for model Mistral-7B.

As for benchmark ALFWorld, we fixed $\alpha = 1$ in all experiments.

Figure 13: An illustrative explanation of our method LAC in BabyAI-Text. The histogram on the right shows the action probabilities of different methods. While actor ($\pi_{LLM}$) and critics (*lang-critic* $\mathcal{C}_{LLM}$, *value-critic* $\mathcal{Q}_{LLM}$) make mistakes at different time steps, LAC (ours) can select the correct action by integrating actor and critics. Please refer to Table 16 for the full trajectory.

Table 3: Results of different hyper-parameter $\alpha$

|  | CodeLlama-7B | Gemma-7B | Llama-3-8B | Mistral-7B |
| --- | --- | --- | --- | --- |
| LAC ($\alpha = 1/2$) | **0.46** | 0.54 | 0.62 | 0.68 |
| LAC ($\alpha = 1$) | **0.46** | 0.62 | 0.64 | 0.58 |
| LAC ($\alpha = 2$) | 0.44 | **0.76** | **0.66** | 0.64 |
| LAC ($\alpha = 5$) | **0.46** | 0.72 | 0.62 | 0.64 |
| LAC ($\alpha = 10$) | 0.40 | 0.58 | 0.60 | **0.70** |

# B  METHOD DETAILS

## B.1  DERIVING THE SOLUTION OF THE KL-CONSTRAINED MAXIMIZATION OBJECTIVE

In this subsection, we will derive Equation (5). We optimize the following objective:

$$\max_{\pi} \mathbb{E}_{a_t^i \sim \pi(a_t^i|g,h_t,c_t)} \left[ \mathcal{Q}_{LLM}(g, h_t, a_t^i, u_t^i) \right] - \frac{1}{\alpha} \mathbb{D}_{KL} \left[ \pi(a_t^i|g, h_t, c_t) \| \pi_{LLM}(a_t^i|g, h_t, c_t) \right].$$

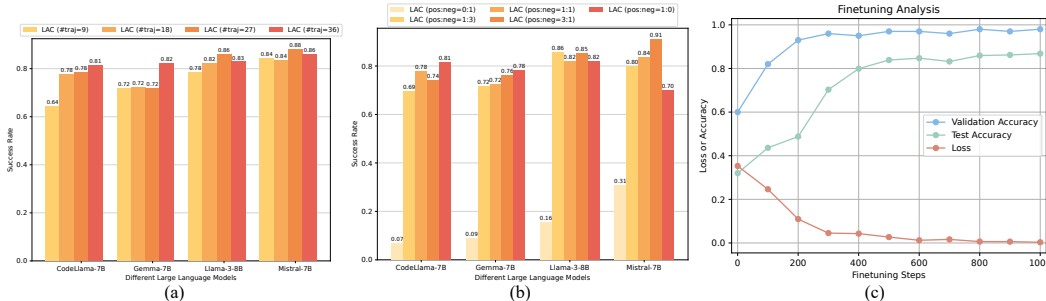

Figure 14: Analysis regarding the fine-tuning process of our LAC. (a) Influence of the fine-tuning data size. Larger data sizes (27, 36 trajectories) generally bring higher performance, but small data sizes (18 and even 9 trajectories) are already enough for our method to achieve outperformance. (b) Influence of the positive/negative data ratio. LAC is robust to reasonable positive/negative ratios (1:3, 1:1, 3:1) while CodeLlama-7B and Gemma-7B-based LAC even perform better given all positive data (1:0). (c) Learning curves of next-token prediction loss and positive/negative tokens prediction accuracy for CodeLlama-7B and ALFWorld.

We now have:

$$\max_{\pi} \mathbb{E}_{a_t^i \sim \pi(a_t^i | g, h_t, c_t)} \left[ \mathcal{Q}_{LLM}(g, h_t, a_t^i, u_t^i) \right] - \frac{1}{\alpha} \mathbb{D}_{KL} \left[ \pi(a_t^i | g, h_t, c_t) \| \pi_{LLM}(a_t^i | g, h_t, c_t) \right]$$

$$= \max_{\pi} \mathbb{E}_{a_t^i \sim \pi(a_t^i | g, h_t, c_t)} \left[ \mathcal{Q}_{LLM}(g, h_t, a_t^i, u_t^i) - \frac{1}{\alpha} \log \frac{\pi(a_t^i | g, h_t, c_t)}{\pi_{LLM}(a_t^i | g, h_t, c_t)} \right]$$

$$= \min_{\pi} \mathbb{E}_{a_t^i \sim \pi(a_t^i | g, h_t, c_t)} \left[ \log \frac{\pi(a_t^i | g, h_t, c_t)}{\pi_{LLM}(a_t^i | g, h_t, c_t)} - \alpha \mathcal{Q}_{LLM}(g, h_t, a_t^i, u_t^i) \right]$$

$$= \min_{\pi} \mathbb{E}_{a_t^i \sim \pi(a_t^i | g, h_t, c_t)} \left[ \log \frac{\pi(a_t^i | g, h_t, c_t)}{\frac{1}{Z(g,h_t,c_t)} \pi_{LLM}(a_t^i | g, h_t, c_t) \exp \left( \alpha \mathcal{Q}_{LLM}(g, h_t, a_t^i, u_t^i) \right)} - \log Z(g, h_t, c_t) \right]$$

where we have the partition function:

$$Z(g, h_t, c_t) = \sum_{a_t^i} \pi_{LLM}(a_t^i | g, h_t, c_t) \exp \left( \alpha \mathcal{Q}_{LLM}(g, h_t, a_t^i, u_t^i) \right).$$

Since the partition function is a function of only $g, h_t$ and the original actor $\pi_{LLM}$, but does not depend on the optimized actor $\pi$, we define

$$\pi^*(a_t^i | g, h_t, c_t) = \frac{1}{Z(g, h_t, c_t)} \pi_{LLM}(a_t^i | g, h_t, c_t) \exp \left( \alpha \mathcal{Q}_{LLM}(g, h_t, a_t^i, u_t^i) \right).$$

This definition of policy if a valid probability distribution as $\pi^*(a_t^i | g, h_t, c_t)$ for all $a_t^i$ and $\sum_{a_t^i} \pi^*(a_t^i | g, h_t, c_t) = 1$. As $Z(g, h_t, c_t)$ is not a function of $a_t^i$, we can then re-organize the objective as:

$$\min_{\pi} \mathbb{E}_{a_t^i \sim \pi(a_t^i | g, h_t, c_t)} \left[ \log \frac{\pi(a_t^i | g, h_t, c_t)}{\frac{1}{Z(g,h_t,c_t)} \pi_{LLM}(a_t^i | g, h_t, c_t) \exp \left( \alpha \mathcal{Q}_{LLM}(g, h_t, a_t^i, u_t^i) \right)} - \log Z(g, h_t, c_t) \right]$$

$$= \min_{\pi} \mathbb{E}_{a_t^i \sim \pi(a_t^i | g, h_t, c_t)} \left[ \log \frac{\pi(a_t^i | g, h_t, c_t)}{\pi^*(a_t^i | g, h_t, c_t)} - \log Z(g, h_t, c_t) \right]$$

$$= \min_{\pi} \mathbb{D}_{KL} \left[ \pi(a_t^i | g, h_t, c_t) \| \pi^*(a_t^i | g, h_t, c_t) \right] - \log Z(g, h_t, c_t).$$

Then since $Z(g, h_t, c_t)$ does not depend on $\pi$, we can only care about the KL-divergence, which is minimized at 0 if and only if the two distributions are identical. Therefore, the optimal solution is

$$\pi(a_t^i | g, h_t, c_t) = \pi^*(a_t^i | g, h_t, c_t) = \frac{1}{Z(g, h_t, c_t)} \pi_{LLM}(a_t^i | g, h_t, c_t) \exp \left( \alpha \mathcal{Q}_{LLM}(g, h_t, a_t^i, u_t^i) \right), \tag{7}$$

which completes the derivation.

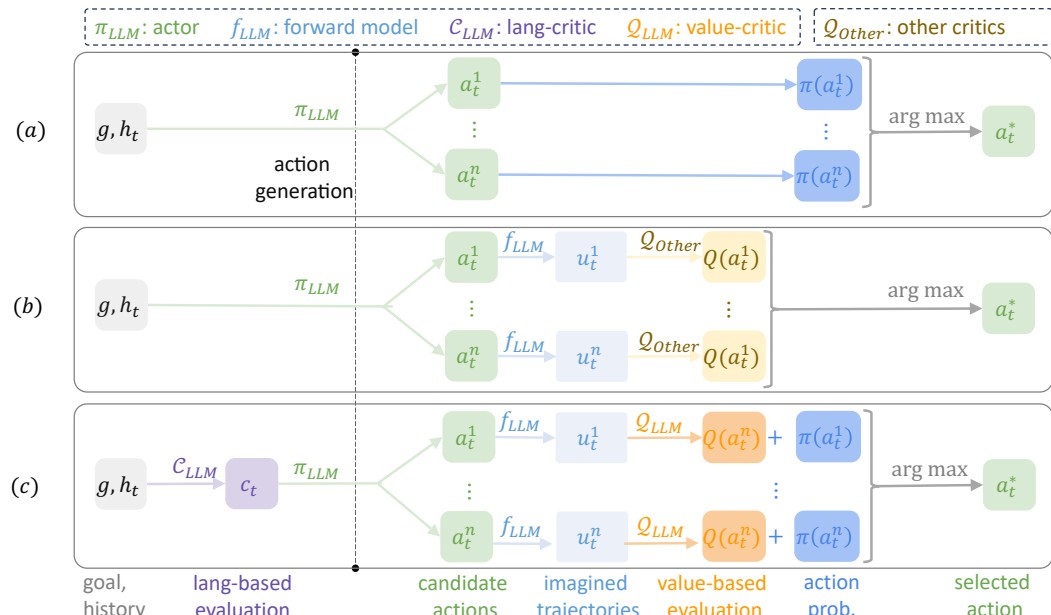

Figure 15: Frameworks comparison. (a) *Actor-only* methods directly select the action with the highest probability generated by actor-LLM $\pi_{LLM}$, which may result in a lack of long-term planning and non-optimal action selection; (b) *Critic-only* methods self-evaluate each candidate action with another critic-LLM $\mathcal{Q}_{LLM}$ by first predicting candidate's future trajectory $u_t^i$ and then directly select the action with the best-predicted outcome, which may ignore the prior knowledge in actor; (c) Our LLM-based Actor-Critic (LAC) algorithm integrate actor and dual critics: *lang-critic* $\mathcal{C}_{LLM}$ and *value-critic* $\mathcal{Q}_{LLM}$ to enhance the decision-making ability of LLMs.

## B.2 CRITIC IMPROVEMENT OF LAC

The *lang-critic* $\mathcal{C}_{LLM}$, *value-critic* $\mathcal{Q}_{LLM}$, and forward model $f_{LLM}$ we used can be easily implemented by prompting LLMs via providing instructions or few-shot examples from similar tasks like prior work (Yao et al., 2022b; Liu et al., 2023). However, empirically, we found that they can be further improved via fine-tuning LLMs with simple next-token prediction loss on several samples collected from training tasks. In this work, we consider 18 trajectories for each benchmark for fine-tuning. Though 18 trajectories are significantly fewer than what is required for conventional reinforcement learning algorithms, they are generally enough for our method. Each trajectory has the following format: $(g, o_0, a_1, o_1, c_1, \cdots, a_H, o_H, c_H)$, where $H$ is the episode length and $c_i$ is a language-based evaluation of action $a_t$. Each $c_t$ includes an explanation about the action $a_t$ (*e.g.*, "I have found object-X. This step is " or "I should take object-X instead of object-Y first. This step is ") and a special token that indicates positive/negative judgment (*e.g.*, "GOOD" or "BAD").

Practically, we just fine-tune the LLM once and use it to construct all the *lang-critic* $\mathcal{C}_{LLM}$, *value-critic* $\mathcal{Q}_{LLM}$, and forward model $f_{LLM}$, thanks to the fine-tuning with the above data format. Specifically, when minimizing the loss of predicting future trajectories, the forward model $f_{LLM}$ is improved. When minimizing the loss of generating language-based evaluations, the *lang-critic* $\mathcal{C}_{LLM}$, *value-critic* $\mathcal{Q}_{LLM}$ are both improved. The latter is because language-based evaluations also contain special tokens that indicate positive/negative judgments, whose generated probabilities are used to calculate $\mathcal{Q}_{LLM}$ in Equation (3). We analyze this fine-tuning process in Appendix A.9. Some examples of the labeled trajectories in ALFWorld and BabyAI-Text are shown in Table 14 and Table 15 respectively.

## B.3 *Actor-only* METHODS

We compare our LAC with actor-only and critic-only methods in Figure 15.

We detail the general *actor-only* methods in Algorithm 2.

---

**Algorithm 2:** *Actor-only* methods.

---

**Input:** current task goal $g$, history $h_t$, actor $\pi_{LLM}$, candidate action size $n$.
**Output:** selected action $a_t^*$

1  $\{a_t^1, a_t^2, \cdots, a_t^n\} \sim \pi_{LLM}(\cdot|g, h_t)$
2  $a_t^* \leftarrow \arg\max_{a_t^i} \pi(a_t^i|g, h_t)$

---

### B.4 *Critic-only* METHODS

We detail the general *critic-only* methods in Algorithm 3. Note that critic-only may use different $\mathcal{Q}_{Other}$ to estimate numerical assessment of actions.

---

**Algorithm 3:** *Critic-only* methods.

---

**Input:** current task goal $g$, history $h_t$, actor $\pi_{LLM}$, forward model $f_{LLM}$, value-based critic $\mathcal{Q}_{Other}$, candidate action size $n$.
**Output:** selected action $a_t^*$

1  $\{a_t^1, a_t^2, \cdots, a_t^n\} \sim \pi_{LLM}(\cdot|g, h_t)$;                   ▷ generate candidate actions
2  **for** $i \leftarrow 1, 2, \cdots, n$ **do**
3  $\quad$ $u_t^i \leftarrow f_{LLM}(g, h_t, a_t^i)$;                   ▷ imagine future trajectory
4  $\quad$ calculate $\mathcal{Q}_{Other}(g, h_t, a_t^i, u_t^i)$
5  **end**
6  $a_t^* \leftarrow \arg\max_{a_t^i} \mathcal{Q}_{Other}(g, h_t, a_t^i, u_t^i)$

---

## C    EXPERIMENT DETAILS

### C.1    BENCHMARK DETAILS

#### C.1.1    ALFWORLD: BENCHMARK WITH HIGH-LEVEL ACTIONS

We choose ALFWorld (Shridhar et al., 2021), a text-based household environment, to demonstrate the effectiveness of `LAC` on high-level planning. ALFWorld is a synthetic text-based game aligned with ALFRED (Shridhar et al., 2020) benchmark. There are 6 types of tasks in this environment, which require the agent to achieve a high-level goal through a sequence of high-level actions, *e.g.* "go to place-X", "take object-Y from place-X", *etc*. The details about the 6 task types in ALFWorld are shown in Table 4.

Table 4: All the task types and the corresponding goals for ALFWorld

| Type | Description |
|---|---|
| Pick & Place | The agent needs to put a target object to a target place, *e.g.* put some spraybottle on toilets, find some apple and put it in sidetable, *etc*. |
| Clean & Place | The agent needs to find a target object, clean it and put it to a target place, *e.g.* clean some apple and put it in sidetable, put a clean lettuce in diningtable, *etc*. |
| Heat & Place | The agent needs to find a target object, heat it and put it to a target place, *e.g.* heat some egg and put it in diningtable, put a hot apple in fridge, *etc*. |
| Cool & Place | The agent needs to find a target object, cool it and put it to a target place, *e.g.* cool some pan and put it in stoveburner, put a cool mug in shelf, *etc*. |
| Examine & Place | The agent needs to find a target object, and examine it with desklamp, *e.g.* look at bowl under the desklamp, examine the pen with the desklamp, *etc*. |
| Pick Two & Place | The agent needs to put two target objects to a target place, *e.g.* put two saltshaker in drawer, find two pen and put them in dresser, *etc*. |

A challenge built into ALFWorld is that the agent needs to explore the environment to find a target object. The commonsense knowledge in LLMs about the likely locations for common household items makes this environment suitable for LLMs to solve. The reward is 1 only when the agent reaches the goal. Following `ReAct`, we evaluate 134 unseen evaluation games in a task-specific setup.

### C.1.2 BABYAI-TEXT: BENCHMARK WITH LOW-LEVEL ACTIONS

For decision-making tasks with low-level planning, we adopt BabyAI-Text (Carta et al., 2023b) as our test-bed. BabyAI-Text is a text-only version environment extended from the BabyAI platform (Chevalier-Boisvert et al., 2019). BabyAI-Text is a Grid World environment, in which the agent and objects are placed in a room of $8 \times 8$ tiles. The agent has 6 primitive actions: turn left, turn right, go forward, pick up, drop, toggle, to solve a task described in natural language (*e.g.* Pick up the red box). The agent has access to a $7 \times 7$ partial view, which means it can only observe the objects belonging to the $7 \times 7$ grid in front of it. In addition to objects relevant to completing a given task, there are also other distractors in the room. All the task types in BabyAI-Text are shown in Table 5.

Table 5: All the task types and the corresponding goals for BabyAI-Text

| Type | Description |
|---|---|
| go to | The agent needs to find target object and go to it, *e.g.* go to the green key, go to the red ball, *etc.* |
| pick up | The agent needs to find target object, go to it and pick up it, *e.g.* pick up the blue key, pick up the purple ball, *etc.* |
| go to after pick up | The agent needs to find and pick up one object, then go to another object, *e.g.* go to the blue key after you pick up the green key *etc.* |
| pick up then go to | The agent needs to find and pick up one object, then go to another object, *e.g.* pick up the green box, then go to the purple box *etc.* |
| put next to | The agent needs to find and pick up one object, then go to another object and put the first object next to it, *e.g.* put the grey key next to the yellow ball *etc.* |
| open door | The agent needs to know which key to pick up, then find and pick up it to open the door, *e.g.* open the door, open the blue door, *etc.* |

Unlike ALFWorld, an agent interacting with BabyAI-Text needs to find out the suitable low-level action to execute at each step. We evaluate on the test environment in BabyAI-Text. The objects in a room are randomly chosen, and their position, as well as the agent's position, are also random. Considering the time and computational resource constraints, we evaluate on 50 tasks for each task type, yielding 300 tasks total.

## C.2 IMPLEMENTATION DETAILS OF BASELINES

### C.2.1 REACT

ReAct (Yao et al., 2022b) is an *actor-only* method that combines reasoning and acting during the interaction with the environment. It leverages the reasoning capabilities of LLMs to increase the probability of the LLM acting correctly as an actor.

Table 6 shows an example of a trajectory interleaving between actions and observations in ALFWorld. To leverage the reasoning ability of LLMs, ReAct adds an extra thinking action to the original action set in ALFWorld, allowing LLM agent to engage in reasoning about task decomposition and goal achievement during task execution. An annotated trajectory for ALFWorld given by ReAct is shown in Table 7. Text labeled in light green is the annotated thought to help LLM complete a task better.

Table 6: An example of successful trajectory for ALFWorld

```
You are in the middle of a room.  Looking quickly around you, you see a cabinet
4, a cabinet 3, a cabinet 2, a cabinet 1, a countertop 1, a garbagecan 1, a
handtowelholder 2, a handtowelholder 1, a sinkbasin 2, a sinkbasin 1, a toilet 1,
a toiletpaperhanger 1, and a towelholder 1.
Your task is to:  put some spraybottle on toilet.
> go to cabinet 1
On the cabinet 1, you see a cloth 1, a soapbar 1, a soapbottle 1.
> go to cabinet 2
The cabinet 2 is closed.
> open cabinet 2
You open the cabinet 2.  The cabinet 2 is open.  In it, you see a candle 1, and a
spraybottle 2.
> take spraybottle 2 from cabinet 2
You pick up the spraybottle 2 from the cabinet 2.
> go to toilet 1
On the toilet 1, you see a soapbottle 2.
> put spraybottle 2 in/on toilet 1
You put the spraybottle 2 in/on the toilet 1.
```

Table 7: An example of annotated successful trajectory for ALFWorld in `ReAct`

```
You are in the middle of a room.  Looking quickly around you, you see a cabinet
4, a cabinet 3, a cabinet 2, a cabinet 1, a countertop 1, a garbagecan 1, a
handtowelholder 2, a handtowelholder 1, a sinkbasin 2, a sinkbasin 1, a toilet 1,
a toiletpaperhanger 1, and a towelholder 1.
Your task is to:  put some spraybottle on toilet.
> think:  To solve the task, I need to find and take a sparybottle, then put it on
toilet.
OK.
> think:  First I need to find a spraybottle.  A spraybottle is more likely to appear
in cabinet (1-4), countertop (1), toilet (1), sinkbasin (1-2), garbagecan (1).  I can
check one by one, starting with cabinet 1.
OK.
> go to cabinet 1
On the cabinet 1, you see a cloth 1, a soapbar 1, a soapbottle 1.
> go to cabinet 2
The cabinet 2 is closed.
> open cabinet 2
You open the cabinet 2.  The cabinet 2 is open.  In it, you see a candle 1, and a
spraybottle 2.
> think:  Now I find a spraybottle (2).  Next, I need to take it.
> take spraybottle 2 from cabinet 2
You pick up the spraybottle 2 from the cabinet 2.
> think:  Now I take a spraybottle (2).  Next, I need to put it in/on toilet 1.
OK.
> go to toilet 1
On the toilet 1, you see a soapbottle 2.
> put spraybottle 2 in/on toilet 1
You put the spraybottle 2 in/on the toilet 1.
```

During the interaction with the environment, two successful trajectories like in Table 7 are provided to LLMs to leverage the in-context learning ability of LLMs. The LLMs will generate an action to be executed in the environment or conduct some reasoning about how to achieve the final goal. The chosen action and new observation are appended to the prompt for LLMs to form a sequential decision-making process.

Table 8: An example of successful trajectory for BabyAI-Text

```
Goal of the agent:  go to the green key
Observation:You see a wall 2 steps right, You see a wall 3 steps forward, You see a
grey box 3 steps left, You see a grey key 3 steps left and 1 step forward, You see a
blue box 3 steps left and 2 steps forward
Action:turn left
Observation:You see a wall 3 steps right, You see a blue key 3 steps left and 2 steps
forward, You see a green key 2 steps left and 1 step forward, You see a green ball
1 step left and 3 steps forward, You see a grey box 3 steps forward, You see a grey
key 1 step right and 3 steps forward, You see a blue box 2 steps right and 3 steps
forward
Action:go forward
Observation:You see a wall 3 steps right, You see a blue key 3 steps left and 1 step
forward, You see a green key 2 steps left, You see a green ball 1 step left and 2
steps forward, You see a grey box 2 steps forward, You see a grey key 1 step right
and 2 steps forward, You see a blue box 2 steps right and 2 steps forward
Action:turn left
Observation:You see a wall 3 steps left, You see a green box 2 steps left and 3 steps
forward, You see a green box 2 steps left and 1 step forward, You see a green key 2
steps forward, You see a blue key 1 step right and 3 steps forward, You see a green
ball 2 steps right and 1 step forward, You see a grey box 2 steps right
Action:go forward
Observation:You see a green box 2 steps left and 2 steps forward, You see a green box
2 steps left, You see a green key 1 step forward, You see a blue key 1 step right and
2 steps forward, You see a green ball 2 steps right
```

Similar to ALFWorld, we annotate some successful trajectories in BabyAI-Text to teach a `ReAct` agent how to complete a task in BabyAI-Text. Examples of successful trajectories without thoughts and with thoughts are shown in Table 8 and Table 9 respectively.

### C.2.2 RAP

RAP(Hao et al., 2023) can be treated as a *critic-only* method, in which it utilizes LLMs as actor and world model, and adopts tree-search planning methods to evaluate each possible action candidate. Trajectories interleaving between actions and observations like in Table 6 and Table 8 are provided

Table 9: An example of annotated successful trajectory for BabyAI-Text in `ReAct`

```
Goal of the agent:  go to the green key
Observation:You see a wall 2 steps right, You see a wall 3 steps forward, You see a
grey box 3 steps left, You see a grey key 3 steps left and 1 step forward, You see a
blue box 3 steps left and 2 steps forward
Action:think:  To complete the task, I need to find a green key and go to it.
Observation:OK
Action:think:  I have not seen a green key.  Firstly, I need to find a green key.
Observation:OK
Action:turn left
Observation:You see a wall 3 steps right, You see a blue key 3 steps left and 2 steps
forward, You see a green key 2 steps left and 1 step forward, You see a green ball
1 step left and 3 steps forward, You see a grey box 3 steps forward, You see a grey
key 1 step right and 3 steps forward, You see a blue box 2 steps right and 3 steps
forward
Action:think:  I have seen a green key 2 steps left and 1 step forward.  Next, I need
to go to the green key.
Observation:OK
Action:go forward
Observation:You see a wall 3 steps right, You see a blue key 3 steps left and 1 step
forward, You see a green key 2 steps left, You see a green ball 1 step left and 2
steps forward, You see a grey box 2 steps forward, You see a grey key 1 step right
and 2 steps forward, You see a blue box 2 steps right and 2 steps forward
Action:turn left
Observation:You see a wall 3 steps left, You see a green box 2 steps left and 3 steps
forward, You see a green box 2 steps left and 1 step forward, You see a green key 2
steps forward, You see a blue key 1 step right and 3 steps forward, You see a green
ball 2 steps right and 1 step forward, You see a grey box 2 steps right
Action:go forward
Observation:You see a green box 2 steps left and 2 steps forward, You see a green box
2 steps left, You see a green key 1 step forward, You see a blue key 1 step right and
2 steps forward, You see a green ball 2 steps right
```

to LLMs to learn how to choose action as an actor and the dynamics of the environment as a world model.

The assessment of each step is performed by a reward function, which can be the log probability of the action or self-evaluation given by LLMs, or based on some task-specific heuristics. In our implementation, we adopt the log probability of actions given by LLMs as the reward. For simplicity of implementation, we adopted a greedy approach to expand the tree, generating only one action at a time. More specifically, at each step, LLMs will sample some action candidates. For each action candidate, LLMs will generate a rollout trajectory until a maximum step or terminal state. The summation of log probabilities of all the actions on the rollout accessed by LLMs are used as Q value for each action candidate. The candidate with the highest Q value is chosen to be executed in the environment.

### C.2.3   ICPI

ICPI (Brooks et al., 2024) proposes to implement policy iteration using LLMs through in-context learning. At each step, the actor in ICPI will sample some action candidates and the critic will compute the Q values for each action candidates. The action candidates with the highest Q values is chosen to be executed.

The actor is implemented using LLMs, and successful trajectories like in Table 6 and Table 8 are provided to it.

As to the critic, ICPI prompts LLMs to give the numerical reward for each step directly. Given the current history and an action candidate, the critic in ICPI will rollout a trajectory starting from the action candidate. Apart from predicting the observations, the critic will give the numerical reward for each step on the rollout trajectory. The (discounted) return on the rollout is treated as the Q value for the action candidate. For both ALFWorld and BabyAI-Text, we define the reward as 1 when the agent reaches the goal. All other steps will have a reward 0. The examples provided to the critic are like in Table 10 for ALFWorld and Table 11 for BabyAI-Text.

Table 10: An example provided to critic in ICPI for ALFWorld

```
You are in the middle of a room.  Looking quickly around you, you see a cabinet
4, a cabinet 3, a cabinet 2, a cabinet 1, a countertop 1, a garbagecan 1, a
handtowelholder 2, a handtowelholder 1, a sinkbasin 2, a sinkbasin 1, a toilet 1,
a toiletpaperhanger 1, and a towelholder 1.
Your task is to:  put some spraybottle on toilet.
> go to cabinet 1
On the cabinet 1, you see a cloth 1, a soapbar 1, a soapbottle 1.
Reward:0
> go to cabinet 2
The cabinet 2 is closed.
Reward:0
> open cabinet 2
You open the cabinet 2.  The cabinet 2 is open.  In it, you see a candle 1, and a
spraybottle 2.
Reward:0
> take spraybottle 2 from cabinet 2
You pick up the spraybottle 2 from the cabinet 2.
Reward:0
> go to toilet 1
On the toilet 1, you see a soapbottle 2.
Reward:0
> put spraybottle 2 in/on toilet 1
You put the spraybottle 2 in/on the toilet 1.
Reward:1
```

Table 11: An example provided to critic in ICPI for BabyAI-Text

```
Goal of the agent:  go to the green key
Observation:You see a wall 2 steps right, You see a wall 3 steps forward, You see a
grey box 3 steps left, You see a grey key 3 steps left and 1 step forward, You see a
blue box 3 steps left and 2 steps forward
Action:turn left
Observation:You see a wall 3 steps right, You see a blue key 3 steps left and 2 steps
forward, You see a green key 2 steps left and 1 step forward, You see a green ball
1 step left and 3 steps forward, You see a grey box 3 steps forward, You see a grey
key 1 step right and 3 steps forward, You see a blue box 2 steps right and 3 steps
forward
Reward:0
Action:go forward
Observation:You see a wall 3 steps right, You see a blue key 3 steps left and 1 step
forward, You see a green key 2 steps left, You see a green ball 1 step left and 2
steps forward, You see a grey box 2 steps forward, You see a grey key 1 step right
and 2 steps forward, You see a blue box 2 steps right and 2 steps forward
Reward:0
Action:turn left
Observation:You see a wall 3 steps left, You see a green box 2 steps left and 3 steps
forward, You see a green box 2 steps left and 1 step forward, You see a green key 2
steps forward, You see a blue key 1 step right and 3 steps forward, You see a green
ball 2 steps right and 1 step forward, You see a grey box 2 steps right
Reward:0
Action:go forward
Observation:You see a green box 2 steps left and 2 steps forward, You see a green box
2 steps left, You see a green key 1 step forward, You see a blue key 1 step right and
2 steps forward, You see a green ball 2 steps right
Reward:1
```

### C.2.4  RAFA

The framework of RAFA (Liu et al., 2023) is also like RAP or ICPI. The main difference is the critic used.

RAFA implements tree-search using LLM to evaluate each action candidate. Different from ICPI, RAFA uses the task completion progress as the value for each step. They have the LLMs decompose a goal into sub-goals, and use the completion status of the sub-goals after each step as the value for the step. RAFA evaluates the completion status of sub-goals based on the predicted observations. Examples provided to critic in RAFA are like in Table 12 for ALFWorld and Table 13 for BabyAI-Text.

Table 12: An example provided to critic in RAFA for ALFWorld

```
You are in the middle of a room.  Looking quickly around you, you see a cabinet
4, a cabinet 3, a cabinet 2, a cabinet 1, a countertop 1, a garbagecan 1, a
handtowelholder 2, a handtowelholder 1, a sinkbasin 2, a sinkbasin 1, a toilet 1,
a toiletpaperhanger 1, and a towelholder 1.
Your task is to:  put some spraybottle on toilet.
> critic:  My task requires two sub-goals in order:  take a spraybottle and put the
spraybottle on the toilet.  My current state satisfies zero of the two sub-goals.
The value is 0/2=0.
> OK.
On the cabinet 1, you see a cloth 1, a soapbar 1, a soapbottle 1.
> OK.
The cabinet 2 is closed.
> OK.
You open the cabinet 2.  The cabinet 2 is open.  In it, you see a candle 1, and a
spraybottle 2.
> OK.
You pick up the spraybottle 2 from the cabinet 2.
> critic:  Now I take a spraybottle.  My current state satisfies the first of the two
sub-goals:  take a spraybottle.  The value is 1/2=0.5.
> OK.
On the toilet 1, you see a soapbottle 2.
> OK.
You put the spraybottle 2 in/on the toilet 1.
> critic:  Now I put the spraybottle on the toilet.  My current state satisfies all
the two sub-goals.  The value is 2/2=1.
```

Table 13: An example provided to critic in RAFA for BabyAI-Text

```
Goal of the agent:  go to the green key
You see a wall 2 steps right, You see a wall 3 steps forward, You see a grey box 3
steps left, You see a grey key 3 steps left and 1 step forward, You see a blue box 3
steps left and 2 steps forward
>critic:  My task requires two sub-goals in order:  find the green key, and go to
the green key.  My current state satisfies zero of the two sub-goals.  The value is
0/2=0.
>OK.
You see a wall 3 steps right, You see a blue key 3 steps left and 2 steps forward,
You see a green key 2 steps left and 1 step forward, You see a green ball 1 step left
and 3 steps forward, You see a grey box 3 steps forward, You see a grey key 1 step
right and 3 steps forward, You see a blue box 2 steps right and 3 steps forward
>critic:  Now I find the green key.  My current state satisfies the first of the two
sub-goals:  find the green key.  The value is 1/2=0.5.
>OK.
You see a wall 3 steps right, You see a blue key 3 steps left and 1 step forward, You
see a green key 2 steps left, You see a green ball 1 step left and 2 steps forward,
You see a grey box 2 steps forward, You see a grey key 1 step right and 2 steps
forward, You see a blue box 2 steps right and 2 steps forward
>OK.
You see a wall 3 steps left, You see a green box 2 steps left and 3 steps forward,
You see a green box 2 steps left and 1 step forward, You see a green key 2 steps
forward, You see a blue key 1 step right and 3 steps forward, You see a green ball
2 steps right and 1 step forward, You see a grey box 2 steps right
>OK.
You see a green box 2 steps left and 2 steps forward, You see a green box 2 steps
left, You see a green key 1 step forward, You see a blue key 1 step right and 2 steps
forward, You see a green ball 2 steps right
>critic:  Now I go to the green key.  My current state satisfies all the two
sub-goals.  The value is 2/2=1.
```

## C.3 IMPLEMENTATION DETAILS OF LAC, ABLATION STUDIES, AND VARIANTS

We build our method based on ReAct (Yao et al., 2022b). Specifically, we adopt the trajectories with thoughts from ReAct and annotate language-based evaluations for those trajectories. For each task type, we annotate two trajectories from ReAct as successful examples to teach LLMs how to complete this type of tasks. Some examples of the annotated trajectories for ALFWorld and BabyAI-Text are shown in Table 14 and Table 15 respectively. The language-based evaluations we annotated are labeled in light orange. We also use ReAct algorithm to collect some failed trajectories from the training set for each task type. The collected failed trajectories are also annotated with language-based evaluations, aiming to teach LLMs to distinguish between steps that are beneficial, harmful, or indeterminate for achieving a goal.

We annotate the language-based evaluations using the following criteria:

Table 14: An example of annotated successful trajectory for ALFWorld in `LAC`

```
You are in the middle of a room.  Looking quickly around you, you see a cabinet
4, a cabinet 3, a cabinet 2, a cabinet 1, a countertop 1, a garbagecan 1, a
handtowelholder 2, a handtowelholder 1, a sinkbasin 2, a sinkbasin 1, a toilet 1,
a toiletpaperhanger 1, and a towelholder 1.
Your task is to:  put some spraybottle on toilet.
> think:  To solve the task, I need to find and take a sparybottle, then put it on
toilet.
OK.
==> critic:  The thought is reasonable.  This step is GOOD.
> think:  First I need to find a spraybottle.  A spraybottle is more likely to appear
in cabinet (1-4), countertop (1), toilet (1), sinkbasin (1-2), garbagecan (1).  I can
check one by one, starting with cabinet 1.
OK.
==> critic:  I need to find spraybottle firstly.  The thought is reasonable.  This
step is GOOD.
> go to cabinet 1
On the cabinet 1, you see a cloth 1, a soapbar 1, a soapbottle 1.
==> critic:  I have not seen spraybottle in this place.  This step is UNKNOWN.
> go to cabinet 2
The cabinet 2 is closed.
==> critic:  The cabinet 2 is closed.  This step is UNKNOWN.
> open cabinet 2
You open the cabinet 2.  The cabinet 2 is open.  In it, you see a candle 1, and a
spraybottle 2.
==> critic:  I have seen spraybottle 2 in this place.  This step is GOOD.
> think:  Now I find a spraybottle (2).  Next, I need to take it.
OK.
==> critic:  I can take the spraybottle after finding it.  The thought is reasonable.
This step is GOOD.
> take spraybottle 2 from cabinet 2
You pick up the spraybottle 2 from the cabinet 2.
==> critic:  I have taken spraybottle 2 in this place.  This step is GOOD.
> think:  Now I take a spraybottle (2).  Next, I need to put it in/on toilet 1.
OK.
==> critic:  I can put the spraybottle in/on toilet after taking it.  The thought is
reasonable.  This step is GOOD.
> go to toilet 1
On the toilet 1, you see a soapbottle 2.
==> critic:  I should go to toilet if I want to put the spraybottle on it.  This step
is GOOD.
> put spraybottle 2 in/on toilet 1
You put the spraybottle 2 in/on the toilet 1.
==> critic:  I have put the spraybottle 2 in/on the toilet 1.  This step is GOOD.
```

(1) If an action on a trajectory is necessary for achieving the final goal, the following language-based evaluations will end with "This step is GOOD." to indicate the completion of an essential step, such as action "take spraybottle 2 from cabinet 2" for the goal "put some spraybottle on toilet".

(2) If an action is crucial for obtaining essential information to complete a goal, the subsequent language-based evaluations will end with "This step is GOOD.", such as action "open cabinet 2" in Table 14 which finds the target object.

(3) If an action is definitely useless or incorrect for achieving a goal, the following language-based evaluations will end with "This step is BAD." to indicate this action is undesirable, such as action "take cloth 1 from cabinet 1" for the task in Table 14.

(4) If an action can not be evaluated as good or bad from the history, the following language-based evaluations will end with "This step is UNKNOWN.", such as action "go to cabinet 1" or "go to cabinet 2" in Table 14.

All the annotated successful and failed trajectories are used to construct fine-tuning examples to fine-tune LLMs to generate better language-based evaluations.

Considering the computational cost of fully finetuning of LLMs, we use LoRA (Hu et al., 2021) to finetune our models. In ALFWorld, with two successful trajectories and one failed trajectory for each task type, we have 485 (input,output) pairs in total of six task types to finetune models. In BabyAI-Text, the number of finetuning (input,output) pairs is 418. We finetune models for 1,000 steps with learning rate 2.5e-5 and batch size 2. We use A100 GPU with 80GB memory to fine-tune our model. With just about 400-500 (input,output) pairs and 1,000 fine-tuning steps, we can complete the fine-tuning process within one and a half hours.

Table 15: An example of annotated successful trajectory for BabyAI-Text in `LAC`

```
Goal of the agent:  go to the green key
Observation:You see a wall 2 steps right, You see a wall 3 steps forward, You see a
critic box 3 steps left, You see a critic key 3 steps left and 1 step forward, You
see a blue box 3 steps left and 2 steps forward
Action:think:  To complete the task, I need to find a green key and go to it.
Observation:OK
Critic:The thought is reasonable.  This step is GOOD.
Action:think:  I have not seen a green key.  Firstly, I need to find a green key.
Observation:OK
Critic:I need to find a green key firstly.  The thought is reasonable.  This step is
GOOD.
Action:turn left
Observation:You see a wall 3 steps right, You see a blue key 3 steps left and 2 steps
forward, You see a green key 2 steps left and 1 step forward, You see a green ball 1
step left and 3 steps forward, You see a critic box 3 steps forward, You see a critic
key 1 step right and 3 steps forward, You see a blue box 2 steps right and 3 steps
forward
Critic:I have turned left and seen a green key 2 steps left and 1 step forward.  This
step is GOOD.
Action:think:  I have seen a green key 2 steps left and 1 step forward.  Next, I need
to go to the green key.
Observation:OK
Critic:I can go to the green key after seeing it.  The thought is reasonable.  This
step is GOOD.
Action:go forward
Observation:You see a wall 3 steps right, You see a blue key 3 steps left and 1 step
forward, You see a green key 2 steps left, You see a green ball 1 step left and 2
steps forward, You see a critic box 2 steps forward, You see a critic key 1 step
right and 2 steps forward, You see a blue box 2 steps right and 2 steps forward
Critic:I have gone forward.  The green key is 2 steps left.  This step is UNKNOWN.
Action:turn left
Observation:You see a wall 3 steps left, You see a green box 2 steps left and 3 steps
forward, You see a green box 2 steps left and 1 step forward, You see a green key 2
steps forward, You see a blue key 1 step right and 3 steps forward, You see a green
ball 2 steps right and 1 step forward, You see a critic box 2 steps right
Critic:I have turned left.  The green key is 2 steps forward.  This step is UNKNOWN.
Action:go forward
Observation:You see a green box 2 steps left and 2 steps forward, You see a green box
2 steps left, You see a green key 1 step forward, You see a blue key 1 step right and
2 steps forward, You see a green ball 2 steps right
Critic:I have gone forward.  The green key is 1 step forward.  This step is GOOD.
```

During testing, the fine-tuned models are used to generate language-based evaluations after executing an action in the environment, as well as to forecast the potential outcomes of each action candidate.

We sample $n = 5$ action candidates from $\pi_{LLM}$ at each time step. The $\pi_{LLM}$ is augmented by leveraging the language-based evaluations generated by $\mathcal{C}_{LLM}$. After sampling action candidates, we use the fine-tuned model to predict future outcomes for each action candidate. The model needs to predict the possible observation and generate language-based evaluations for each predicted step. We set the maximum prediction step as 4, the model will continue the prediction until it generates a language-based evaluation ending with "This step is GOOD." or "This step is BAD", or when it reaches the maximum prediction step.

For the optimization of $\pi_{LLM}$, we solve an optimization problem in Equation (4) with a hyper-parameter $\alpha$, which balances the generating probabilities of $\pi_{LLM}$ and the values given by $\mathcal{Q}_{LLM}$. For ALFWorld, we set $\alpha$ as 1, which yields superb performance over baselines. For BabyAI-Text, we conduct a grid-search over {1/2,1,2,5,10} for $\alpha$, finding that different LLMs will have best performance with different $\alpha$. The results can be seen in Table 3.

We set the maximum horizon length to 40 for ALFWorld and 30 to BabyAI-Text. If the agent has not reached the final goal after 40 or 30 steps, this episode will be marked as failure.

We use A100 GPU with 80GB memory to evaluate our method. For `LAC`, the execution time for ALFWorld is about 10 hours for 134 tasks using single A100 GPU. And for BabyAI-Text, the execution time can be varied for different task types, ranging from 4 to 10 hours for 50 tasks using one A100 GPU. The GPU memory usage may range from 15GB to over 70GB during the interaction according to the length of inputs to LLMs.

We compare our method with all the aforementioned baselines, demonstrating the effectiveness of our method on decision-making tasks with both high-level actions and low-level actions. To demonstrate the effectiveness of each component in our method, we conduct ablation studies on each component. We removes the lang-critic $\mathcal{C}_{LLM}$ from LAC as well as the integration during pre-action-generation phase. This variant is called LAC *w/o* lang-critic. We also evaluate the role of $\mathcal{Q}_{LLM}$ by removing it from LAC as well as the integration during post-action-generation phase. This variant is called LAC *w/o* value-critic. We also demonstrate the role of the action prior given by LLM policy by using only value-critic $\mathcal{Q}_{LLM}$ for decision-making. We call this variant as Value-critic-only. The execution time of those variants during evaluation can be varied according to its performance because a method having poor performance typically will cost more time to execute. On ALFWorld, it may be 10-20 hours. The comparisons between those variants are shown in Figure 4.

We found that each component in LAC is crucial for the superb performance. Removing some components may lead to wrong choice of action candidates. Such an example is shown in Table 16. LAC can complete this task successfully, while eliminating some components in LAC will lead to failure. The comparison is shown in Figure 1.

## D    LIMITATIONS

Our work has two limitations. Firstly, *lang-critic* of LAC is only used before action generation and it can also be applied after action generation. For example, it can provide language-based evaluations for predicted future trajectories to re-sample candidate actions if the previous candidate actions all fail to complete the target task. Secondly, though the *value-critic* of LAC can also adopt tree-search to provide a more accurate assessment of candidate actions, in this paper, we only expand one node for each candidate action for simplicity.

## E    BROADER IMPACTS

Our method is built upon open-source large language models (LLMs). Like other methods that use LLMs, our method also inherits some benefits and challenges from LLMs. For the benefits, our method directly exploits the prior knowledge from LLMs, which may reduce potential carbon costs compared with training policies from scratch. For the challenges, our method might be susceptible to producing unintended output when confronted with harmful input, such as unethical text or input intended for adversarial attacks. To solve this problem, we suggest a thoughtful deployment of our method, such as adding a filtering component.

Table 16: An example of successful trajectory in test environment in BabyAI-Text given by `LAC`

```
Goal of the agent:go to the green ball
Observation:You see a wall 5 steps forward, You see a wall 2 steps left, You see a
yellow ball 1 step left and 4 steps forward, You see a blue ball 1 step left and
3 steps forward, You see a red ball 1 step left and 1 step forward, You see a grey
key 2 steps right and 2 steps forward, You see a green key 2 steps right and 1 step
forward, You see a blue key 3 steps right and 1 step forward
Action:think:To complete the task, I need to find a green ball and go to it.
Observation:OK
Critic:This thought is reasonable.  This step is GOOD.
Action:think:I have seen a green ball 1 step left and 4 steps forward.  Next, I need
to go to the green ball.
Observation:OK
Critic:This thought is reasonable.  I can go to the green ball after seeing it.  This
step is GOOD.
Action:turn right
Observation:You see a wall 5 steps forward, You see a wall 2 steps right, You see a
green ball 3 steps left and 4 steps forward, You see a grey key 2 steps left and 2
steps forward, You see a blue key 1 step left and 3 steps forward, You see a green
key 1 step left and 2 steps forward, You see a blue ball 1 step right and 1 step
forward
Critic:I have turned right.  The green ball is 3 steps left and 4 steps forward.
This step is UNKNOWN.
Action:go forward
Observation:You see a wall 4 steps forward, You see a green ball 3 steps left and 3
steps forward, You see a grey key 2 steps left and 1 step forward, You see a blue key
1 step left and 2 steps forward, You see a green key 1 step left and 1 step forward,
You see a blue ball 1 step right
Critic:I have gone forward.  The green ball is 3 steps left and 3 steps forward.
This step is UNKNOWN.
Action:go forward
Observation:You see a wall 3 steps forward, You see a wall 2 steps right, You see a
green ball 3 steps left and 2 steps forward, You see a grey key 2 steps left, You see
a blue key 1 step left and 1 step forward, You see a green key 1 step left
Critic:I have gone forward.  The green ball is 3 steps left and 2 steps forward.
This step is UNKNOWN.
Action:go forward
Observation:You see a wall 2 steps forward, You see a wall 2 steps right, You see a
green ball 3 steps left and 1 step forward, You see a blue key 1 step left
Critic:But I have gone forward.  The green ball is 3 steps left and 1 step forward.
This step is UNKNOWN.
Action:go forward
Observation:You see a wall 1 step forward, You see a wall 2 steps right, You see a
green ball 3 steps left
Critic:But I have gone forward.  The green ball is 3 steps left.  This step is
UNKNOWN.
Action:turn left
Observation:You see a wall 1 step right, You see a grey key 2 steps left and 2 steps
forward, You see a green key 2 steps left and 1 step forward, You see a blue key 1
step left and 1 step forward, You see a red ball 4 steps forward, You see a green
ball 3 steps forward
Critic:But I have turned left.  The green ball is 3 steps forward.  This step is
UNKNOWN.
Action:go forward
Observation:You see a wall 1 step right, You see a grey key 2 steps left and 1 step
forward, You see a green key 2 steps left, You see a blue key 1 step left, You see a
red ball 3 steps forward, You see a green ball 2 steps forward
Critic:But I have gone forward.  The green ball is 2 steps forward.  This step is
UNKNOWN.
Action:go forward
Observation:You see a wall 1 step right, You see a grey key 2 steps left, You see a
red ball 2 steps forward, You see a green ball 1 step forward
Critic:But I have gone forward.  The green ball is 1 step forward.  This step is
GOOD.
```

