# OpenReview forum: "Enhancing Decision-Making of Large Language Models via Actor-Critic"
_ICLR.cc/2025/Conference — Submitted to ICLR 2025_

### Official Review · Reviewer_Xk7a · 2024-10-23

**Soundness:** 2
**Presentation:** 1
**Contribution:** 2
**Rating:** 3
**Confidence:** 4

**Summary:**

This paper introduces LLM-based Actor Critic framework (LAC) to improve decision-making capabilities of LLM agents through an integration of the actor and the critic. LAC makes use of two different critics including a language critic that provides contextual information and a value critic that provides more quantitative information. The paper also proposed a gradient-free policy improvement approach using two critics without incurring costly backpropagation processes. The effectiveness of LAC is demonstrated in Alfworld and BabyAI-test, and even surpasses GPT4 with ReAct.

**Strengths:**

This paper addresses an important and relevant problem of improving decision-making capabilities of LLM agents.

It is nice that policy improvement can be achieved without incurring costly gradient updates and loss backpropagation.

The paper shows improvements on two popular benchmarks including Alfworld and BabyAI.

**Weaknesses:**

The motivation that "these methods typically adopt either an actor only or critic only approach" (line 42-43) misses many related works. The paper relies on [1] to be the only paper that discusses actor critic methods for LLM agents but many important related works are missing. Even PPO is commonly considered as an actor-critic method where it has an actor that estimates the V function to reduce variance for policy gradient estimation. Thus, many prior works that use PPO for LLM agents should be considered as actor-critic methods (e.g. [2]). Retroformer [3] can also be considered to be an actor-critic method where the critic is a natural language based critic. Other works also applied value-based actor critic methods to LLM agent tasks (e.gl [4]).

The novelty of the method is limited. The language critic and value critic are two main proposals in this paper. However, the language critic is relatively simple and can be considered as a direct use of CoT [5] where the agent is asked to generate thoughts reflecting on the previous round actions before taking actions. The objective of the value critic is also similar to constrained decoding [6] that has been widely used in the alignment domain without the need of performing gradient updates on models.

The writing of the paper, and in particular the motivation (see above) and the experiment section can be improved. Section 5.2 and 5.3 only state the the proposed methods are better than baselines and other ablations without investigating into the reason of the gap. E.g. Why is LAC so much better than ReAct/ RAP, and is there any finding of comparing the performance of LAC with different base models. The experiment section does not provide such necessary analysis information.

It is unclear how generalizable and computationally efficient the proposed method is. In particular, it seems that the method can only be applied to tasks with a finite action space and it is unclear if the method can generalize to realistic tasks with unbounded action space such as web browsing.

The tasks used in this work are more on the simple side. It would be interesting to see if the proposed method can work in more challenging tasks such as web browsing, coding, minecraft etc.

[1] Controlling large language model-based agents for large-scale decision-making: An actor-critic approach
[2] Large language models as generalizable policies for embodied tasks
[3] RETROFORMER: RETROSPECTIVE LARGE LANGUAGE AGENTS WITH POLICY GRADIENT OPTIMIZATION
[4] ArCHer: Training Language Model Agents via Hierarchical Multi-Turn RL
[5] Chain-of-Thought Prompting Elicits Reasoning in Large Language Models
[6] Controlled Decoding from Language Models

**Questions:**

37: long-time horizon -> long horizons

151-152: "Due to the auto-regressive nature of LLM, it does not do reasoning and planning explicitly." This seems controversial. Chain-of-thought/o-1 are also auto-regressive decoding, but arguably they have some reasoning in them. Same with 152-154, "Accordingly, LLM
with actor-only methods often struggles with complex tasks that require multiple steps of planning and reasoning".

Please see weakness.

---

> ### Author Response · Authors · 2024-11-25
> **Additional Experimental Results and Clarifications to Other Questions (Part 1)**
>
> Thanks for the valuable comments and helpful suggestions. Here we provide additional experimental results and detailed explanations for clarifying your questions. The detailed experimental results are provided in the revised version of the paper.
>
> > **Weakness 1**:  The motivation that "these methods typically adopt either an actor only or critic only approach" (line 42-43) misses many related works. The paper relies on [1] to be the only paper that discusses actor-critic methods for LLM agents but many important related works are missing. Even PPO is commonly considered as an actor-critic method where it has an actor that estimates the V function to reduce variance for policy gradient estimation. Thus, many prior works that use PPO for LLM agents should be considered as actor-critic methods (e.g. [2]). Retroformer [3] can also be considered to be an actor-critic method where the critic is a natural language-based critic. Other works also applied value-based actor-critic methods to LLM agent tasks (e.g. [4]).
>
> **A**: Thank you for your insightful feedback. However, our paper primarily addresses enhancing decision-making performance in LLMs with few samples, a context that often contrasts with traditional actor-critic with LLM methods that require large datasets [2,3,4]. We have also recognized the importance of discussing traditional actor-critic with LLM methods and included several works [3,8,9] in the Related Work of our initial submission. We thank your suggestions and will incorporate citations [2,4] to provide a more comprehensive overview of actor-critic methods in the context of LLMs in the revised version of the paper.

---

> ### Author Response · Authors · 2024-11-25
> **Additional Experimental Results and Clarifications to Other Questions (Part 2)**
>
> > **Weakness 2**:  The novelty of the method is limited. The language critic and value critic are two main proposals in this paper. However, the language critic is relatively simple and can be considered as a direct use of CoT [5] where the agent is asked to generate thoughts reflecting on the previous round actions before taking actions. The objective of the value critic is also similar to constrained decoding [6] that has been widely used in the alignment domain without the need of performing gradient updates on models.
>
> **A**: We respectfully disagree with the assessment regarding the novelty of our contributions. Below we address the points raised and clarify the distinct contributions of our work relative to the cited methods.
>
>  - **Novelty of the Language Critic**: (1) While it is true that our language critic draws inspiration from techniques like Chain-of-Thought (CoT) prompting [5], our approach goes beyond merely generating intermediate reflections. Specifically, the language critic in our framework evaluates prior actions using natural language feedback and dynamically integrates this feedback into the actor’s decision-making process through in-context learning. This enables the agent to not only reflect on previous decisions but also refine its subsequent action space in a structured manner. (2) Unlike CoT, which focuses on reasoning for static problem-solving, our language critic is explicitly designed for sequential decision-making tasks, where the evaluation of actions must adapt to the evolving context of the task. This extension is critical in domains requiring iterative feedback loops and real-time adjustments, as demonstrated in our benchmarks (Section 5.2 and 5.3 of the paper).
>  - **Novelty of the Value Critic**: (1) While constrained decoding [6] provides a mechanism for aligning generated outputs with desired constraints, our value critic operates differently in both its objectives and implementation. Constrained decoding typically enforces constraints at the output level, often without evaluating long-term sequential outcomes. In contrast, our value critic explicitly estimates the expected cumulative rewards of actions by leveraging LLMs' internal belief probabilities, coupled with a gradient-free optimization framework. (2) The introduction of trajectory rollouts within our value critic further distinguishes it from constrained decoding. By simulating future outcomes of candidate actions, our approach provides a robust mechanism to estimate long-term action-value relationships, ensuring both contextual alignment and quantitative optimization. As shown in the ablation studies (Section 5.3), this integration leads to a significant performance boost over value-only or heuristic methods.
>  - **Integration of Dual Critics with Policy Improvement**: The most significant novelty of our framework lies in the synergistic integration of the language critic and value critic and enabling gradient-free policy improvement. While each critic independently enhances decision-making, their combined use enables a holistic evaluation of actions, balancing qualitative insights (language critic) with quantitative optimization (value critic). This dual-feedback mechanism is unique to our framework and has not been addressed in prior works such as CoT or constrained decoding.
>
> We believe that these distinctions substantiate the novelty of our proposed methods and their significant contributions to LLM-based sequential decision-making. Thank you for pointing out this opportunity to clarify our contributions.

---

> ### Author Response · Authors · 2024-11-25
> **Additional Experimental Results and Clarifications to Other Questions (Part 3)**
>
> > **Weakness 3**:  The writing of the paper, and in particular the motivation (see above) and the experiment section can be improved. Section 5.2 and 5.3 only state the the proposed methods are better than baselines and other ablations without investigating into the reason of the gap. E.g. Why is LAC so much better than ReAct/ RAP, and is there any finding of comparing the performance of LAC with different base models. The experiment section does not provide such necessary analysis information.
>
> **A**: We thank the reviewer for the suggestions and we add more discussions to explain the performance gaps.
>
> - **LAC vs. ReAct/RAP**: LAC's superior performance stems from its balanced integration of the strengths of both actor and critic. While actor-only (e.g., ReAct) methods excel in short-term actions, they often struggle with long-term reasoning. In contrast, critic-only (e.g., RAP) methods conduct explicit reasoning but might mispredict future trajectories and lead to even worse action selection occasionally compared with actor-only methods. LAC addresses these limitations by balancing the actor's action generation and the critic's evaluation. We have provided illustrative examples for AlfWorld and BabyAI in Figure 1 and Figure 13 respectively. In summary, actor-only and critic-only methods make mistakes at different time steps, our LAC can select the correct action.
>
> - **LAC with Different Base Models**: Regarding the performance of LAC with different base models, we highlight two key findings: (1) Our method is general and can be adapted to various base models, and (2) stronger base models, such as Gemma-7B, demonstrate higher performance when integrated with our approach. However, due to the incomplete public availability of training details for these base models, further in-depth analysis will require additional investigation.
>
> We will include these discussions in our paper to improve the overall clarity and rigor of the paper. Thank you for highlighting these areas for improvement.
>
> > **Weakness 4 & Weakness 5**: (W4:) It is unclear how generalizable and computationally efficient the proposed method is. In particular, it seems that the method can only be applied to tasks with a finite action space and it is unclear if the method can generalize to realistic tasks with unbounded action space such as web browsing. (W5:) The tasks used in this work are more on the simple side. It would be interesting to see if the proposed method can work in more challenging tasks such as web browsing, coding, minecraft etc.
>
> **A**: We have systematically exhibited the computational cost of our method and baselines in Figure 5 of the paper. In brief, although our method does not have the lowest cost per task, when considering both the success rate and the cost, our approach is the most cost-effective.
>
> To show the generalizability of our method, we have conducted new experiments using the WebShop benchmark [10], which simulates a web browsing task with a potentially infinite action space. In this scenario, an agent is required to purchase a product based on specific instructions (e.g. "I need a long clip-in hair extension which is natural looking, and price lower than 20.00 dollars") through web interactions (e.g. search "long clip-in hair extension", click buttons such as "[item ID]" or "back to search"). Within this context, the 'search' and 'click' action can indeed lead to an unbounded set of potential actions, as the agent can continuously refine its queries and selections based on dynamic web results.
>
> We represent the detailed results in Figure 7 of the revised paper, and we also show some results in Table 1 and Table 2 below.  We found that our method, LAC, consistently outperforms other baselines, in terms of both success rate and final reward across various base models. This demonstrates the robustness of our method in handling more complex and unbounded action spaces.
>
> **Table 1: Success rate comparison in WebShop benchmark**
> | Success Rate | CodeLlama-7B | Gemma-7B | Llama-3-8B | Mistral-7B |
> | ------------ | ------------ | -------- | ---------- | ---------- |
> | LAC (ours)   | **32%**      | **46%**  | **39%**    | **32%**    |
> | ReAct        | 15%          | 35%      | 37%        | 30%        |
> | RAP          | 19%          | 28%      | 28%        | 26%        |
>
> **Table 2: Final reward comparison in WebShop benchmark**
> | Reward     | CodeLlama-7B | Gemma-7B   | Llama-3-8B | Mistral-7B |
> | ---------- | ------------ | ---------- | ---------- | ---------- |
> | LAC (ours) | **0.5840**   | **0.7237** | **0.6733** | **0.6299** |
> | ReAct      | 0.5042       | 0.6332     | 0.6445     | 0.6159     |
> | RAP        | 0.5545       | 0.6048     | 0.6215     | 0.5594     |

---

> ### Author Response · Authors · 2024-11-25
> **Additional Experimental Results and Clarifications to Other Questions (Part 4)**
>
> > **Question 1**:  151-152: "Due to the auto-regressive nature of LLM, it does not do reasoning and planning explicitly." This seems controversial. Chain-of-thought/o-1 are also auto-regressive decoding, but arguably they have some reasoning in them. Same with 152-154, "Accordingly, LLM with actor-only methods often struggles with complex tasks that require multiple steps of planning and reasoning".
>
> **A**: There might be some misunderstandings. Our statement refers specifically to the actor-only method, which directly outputs selected actions ($a\sim \pi_{LLM}(\cdot|g,h)$ as defined in the Preliminary Section). This method does not involve explicit reasoning or planning. In contrast, chain-of-thought and o1 incorporate reasoning processes, which distinguishes them from the actor-only method.
>
> Thanks again for your efforts and insightful comments! We hope our clarification addresses your concerns and sincerely appreciate it if you could re-evaluate our work. Any further feedback and discussions are much appreciated.
>
> ---
>
> **References**
>
> [1] Zhang, Bin, et al. "Controlling Large Language Model-based Agents for Large-Scale Decision-Making: An Actor-Critic Approach." ICLR 2024 Workshop on Large Language Model (LLM) Agents.
>
> [2] Szot, Andrew, et al. "Large language models as generalizable policies for embodied tasks." The Twelfth International Conference on Learning Representations. 2023.
>
> [3] Yao, Weiran, et al. "Retroformer: Retrospective Large Language Agents with Policy Gradient Optimization." The Twelfth International Conference on Learning Representations.
>
> [4] Zhou, Yifei, et al. "ArCHer: Training Language Model Agents via Hierarchical Multi-Turn RL." Forty-first International Conference on Machine Learning.
>
> [5] Wei, Jason, et al. "Chain-of-thought prompting elicits reasoning in large language models." Advances in neural information processing systems 35 (2022): 24824-24837.
>
> [6] Mudgal, Sidharth, et al. "Controlled Decoding from Language Models." Forty-first International Conference on Machine Learning.
>
> [7] Rafailov, Rafael, et al. "Direct preference optimization: Your language model is secretly a reward model." Advances in Neural Information Processing Systems 36 (2024).
>
> [8] Carta, Thomas, et al. "Grounding large language models in interactive environments with online reinforcement learning." International Conference on Machine Learning. PMLR, 2023.
>
> [9] Tan, Weihao, et al. "True Knowledge Comes from Practice: Aligning Large Language Models with Embodied Environments via Reinforcement Learning." The Twelfth International Conference on Learning Representations.
>
> [10] Yao, Shunyu, et al. "Webshop: Towards scalable real-world web interaction with grounded language agents." Advances in Neural Information Processing Systems 35 (2022): 20744-20757.
>
> [11] Hao, Shibo, et al. "Reasoning with Language Model is Planning with World Model." NeurIPS 2023 Workshop on Generalization in Planning.
>
> [12] Yao, Shunyu, et al. "ReAct: Synergizing Reasoning and Acting in Language Models." The Eleventh International Conference on Learning Representations.

---

> ### Author Response · Authors · 2024-11-26
> **Follow-Up on Rebuttal for Your Review**
>
> Dear Reviewer,
>
> Thank you for your time and effort in reviewing our work. We have provided detailed clarifications and experimental results in our rebuttal to address the issues and concerns raised in your comments.
>
> If our response satisfactorily resolves your concerns, we kindly ask if you could reconsider your evaluation of our work. Should you have any additional questions or comments, we would be happy to engage in further discussions to ensure all aspects are addressed.
>
> Thank you again for your thoughtful review and support.
>
> Best regards,
> The Authors

---

> > ### Comment · Reviewer_Xk7a · 2024-11-27
> > **Thanks for your rebuttal**
> >
> > Thank the authors for their rebuttal. My following concerns still remain:
> >
> > ### The novelty of language and value critic compared to prior works
> > It seems from the response of the author that the main difference of the language critic from naive chain-of-thought is that naive chain-of-thought reflects on the current state while the language critic reflects on the previous state and action results. It does not seem to make enough contributions as it seems that the only difference is whether the observation space contains the previous states and actions or not. For the value critic, it seems that the main difference from constrained decoding is that the value critic uses Monte Carlo estimations of the Q function instead of actually training a parametric Q function. This practice is also commonly adopted when it is unreliable to train a Q function (e.g. https://arxiv.org/pdf/2406.14532). It is unclear to me what the challenges are to combine those two critics, and simply combining two common practices is not novel enough for a paper that claims to come up with a new method.
> >
> > ### The scope of experiments
> > While the novelty is limited, the domains where experiments are conducted in this paper seem to be limited too. Alfworld and babyAI are no longer the most exciting applications where the state-of-the-art models are competing on. Even WebShop is on the more contrived side compared to more recent benchmarks, where each task can be completed by a sequence of searching, choosing an object, and clicking [buy now]. Since this paper does not perform gradient-based optimizations, it might worth showing impressive results in more realistic and up-to-date benchmarks such as WebArena (https://arxiv.org/abs/2307.13854) and OS World (https://arxiv.org/abs/2404.07972). For the level of complexity of the method proposed, more impressive results are expected on more challenging benchmarks.
> >
> > ### The computation analysis experiment misses important details
> > While the authors present additional results of computation analysis in Section 5.4, it seems to be a red flag to me as many important details are missing. E.g. What is the base model used here, are they the same for all methods? What is the hardware for this to be tested on, is there any necessary parallelism? How is each metric calculated? Why does ReAct have more steps but much less time per task? Does "steps per task" take into account the costs of estimating Q(s,a) with trajectory rollouts?
> >
> > Therefore, I maintain my original score and suggest that the paper would benefit from more explanations on the challenges of combining language and value critic, more experiments on most up-to-date challenging benchmarks, and more clarifications and re-writing of the experiments section.

---

> > > ### Author Response · Authors · 2024-11-30
> > > **New Experimental Results and Clarifications to Other Questions (Part 1)**
> > >
> > > ### **Question 1:**  The novelty of language and value critic compared to prior works.
> > >
> > > > **Question 1.1:** It seems from the response of the author that the main difference of the language critic from naive chain-of-thought is that naive chain-of-thought reflects on the current state while the language critic reflects on the previous state and action results. It does not seem to make enough contributions as it seems that the only difference is whether the observation space contains the previous states and actions or not.
> > >
> > > **A**: We respectfully disagree with the claim about the novelty of lang-critic. Though the lang-critic draws inspiration from CoT, it is not trivial to determine what the reflections should be. The lang-critic generates judgments based on previous actions and their outcomes, whereas CoT uses arbitrary thought generation without a structured focus on past mistakes. The lang-critic is specially designed for decision-making tasks and owns two advantages over naive CoT:
> > >
> > > (1) Lang-critic can improve the policy directly by avoiding previous mistakes when the actor is conditioned on the lang-critic's generation.
> > >
> > > To substantiate this claim, we conducted experiments comparing the performance of `actor w/ lang-critic` and `actor w/ CoT` on the WebShop benchmark. Here are the details of these two variants:
> > >
> > > - `actor w/ lang-critic`: We remove all components of LAC, leaving the actor and lang-critic unchanged. Specifically, at each step, after observing the action results, the lang-critic first generates some judgments on previous actions, and then the actor selects the next action based on the judgments.
> > > - `actor w/ CoT`: We remove all components of LAC, except the actor. Additionally, we equip the actor with CoT by adding "Let's think step by step" to the prompt. Specifically, at each step, before choosing the next action, the CoT prompting component first outputs arbitrary thoughts that may help solve the task.
> > >
> > > As shown in Table 3 and Table 4 below, `actor w/ lang-critic` consistently surpasses `actor w/ CoT` across most base models in terms of both Success rate and Reward. By analyzing the results, we found that `actor w/ CoT` may make the same mistake multiple times and get stuck at this mistake, while `actor w/ lang-critic` can largely avoid seen mistakes.
> > >
> > > **Table 3: Success rate comparison of `actor w/ lang-critic` and `actor w/ CoT` in WebShop benchmark**
> > >
> > > | Success Rate         | CodeLlama-7B | Gemma-7B | Llama-3-8B | Mistral-7B |
> > > | -------------------- | ------------ | -------- | ---------- | ---------- |
> > > | actor w/ lang-critic | **21%**      | **46%**  | **39%**    | **31%**    |
> > > | actor w/ CoT         | **21%**      | 20%      | 31%        | 20%        |
> > >
> > >
> > > **Table 4: Final reward comparison of `actor w/ lang-critic` and `actor w/ CoT` in WebShop benchmark**
> > >
> > > | Reward               | CodeLlama-7B | Gemma-7B   | Llama-3-8B | Mistral-7B |
> > > | -------------------- | ------------ | ---------- | ---------- | ---------- |
> > > | actor w/ lang-critic | **0.5739**   | **0.6564** | **0.6556** | **0.6288** |
> > > | actor w/ CoT         | 0.5520       | 0.5347     | 0.6379     | 0.4671     |
> > >
> > >
> > > (2) Lang-critic can be seamlessly integrated with value-critic, which helps value-critic to generate more accurate value-based evaluations.
> > >
> > > To further evaluate this integration, we compare the performance of `LAC` and `LAC w/ CoT` on the WebShop benchmark. The details of the two methods are as follows:
> > >
> > > - `LAC`: Our original method.
> > > - `LAC w/ CoT`: We replace the lang-critic component of LAC with CoT and keep other components unchanged.
> > >
> > > We show the results in Table 5 and Table 6 below. Our method `LAC` consistently outperforms `LAC w/ CoT` regarding Success rate and Reward across all evaluated base models. This is because, without lang-critic's judgment on previous steps, the value-critic may output inaccurate value-based estimations, hindering the policy improvement phase.
> > >
> > > **Table 5: Success rate comparison of `LAC` and `LAC w/ CoT` in WebShop benchmark**
> > >
> > > | Success Rate | CodeLlama-7B | Gemma-7B | Llama-3-8B | Mistral-7B |
> > > | ------------ | ------------ | -------- | ---------- | ---------- |
> > > | LAC (ours)   | **32%**      | **46%**  | **39%**    | **32%**    |
> > > | LAC w/ CoT   | 27%          | 29%      | 33%        | 24%        |
> > >
> > > **Table 6: Final reward comparison of `LAC` and `LAC w/ CoT` in WebShop benchmark**
> > >
> > > | Reward     | CodeLlama-7B | Gemma-7B   | Llama-3-8B | Mistral-7B |
> > > | ---------- | ------------ | ---------- | ---------- | ---------- |
> > > | LAC (ours) | **0.5840**   | **0.7237** | **0.6733** | **0.6299** |
> > > | LAC w/ CoT | 0.5734       | 0.5270     | 0.6515     | 0.5101     |
> > >
> > > In summary, the lang-critic's focused evaluations of past actions provide critical advantages over the simple reflections of CoT, leading to improved performance in decision-making tasks.

---

> > > ### Author Response · Authors · 2024-11-30
> > > **New Experimental Results and Clarifications to Other Questions (Part 3)**
> > >
> > > ### **Question 2:** The scope of experiments.
> > >
> > > > **Question 2.1:** While the novelty is limited, the domains where experiments are conducted in this paper seem to be limited too. Alfworld and babyAI are no longer the most exciting applications where the state-of-the-art models are competing on. Even WebShop is on the more contrived side compared to more recent benchmarks, where each task can be completed by a sequence of searching, choosing an object, and clicking [buy now].
> > >
> > > **A**: We respectively disagree with this claim.
> > >
> > > (1) AlfWorld and BabyAI remain widely used benchmarks in the research community, as evidenced by their inclusion in multiple recent studies [6,7,8,9]. These environments provide a valuable foundation for evaluating the capabilities of state-of-the-art LLM agent methods in simulated settings that mimic real-world household decision-making processes.
> > >
> > > (2) Although the WebShop environment may appear straightforward for humans, it presents significant challenges for state-of-the-art LLM agents. As indicated in Table 9 below, our method (LAC) outperforms several existing approaches, yet it still falls short of human-level performance:
> > >
> > > **Table 9: SOTA performance on WebShop benchmark**
> > >
> > > |                                 | Success Rate | Reward     |
> > > | ------------------------------- | ------------ | ---------- |
> > > | LAC (ours), Gemma-7B            | 46%          | 0.7237     |
> > > | LATS [10], Gemma-7B             | 39%          | 0.6313     |
> > > | LATS [10], GPT-3.5              | 38%          | 0.7590     |
> > > | Auto-GPT [11], GPT-3.5          | 23%          | 0.5282     |
> > > | Auto-GPT [11], GPT-4            | 32%          | 0.6155     |
> > > | Expert Human, results from [10] | **59.6%**    | **0.8210** |
> > >
> > >
> > > > **Question 2.2:** Since this paper does not perform gradient-based optimizations, it might worth showing impressive results in more realistic and up-to-date benchmarks such as WebArena ([2] https://arxiv.org/abs/2307.13854) and OS World ([3] https://arxiv.org/abs/2404.07972). For the level of complexity of the method proposed, more impressive results are expected on more challenging benchmarks.
> > >
> > > **A**: Thank you for your valuable suggestion regarding the use of advanced benchmarks such as WebArena [2] and OS World [3]. While these benchmarks are indeed challenging and valuable for assessing complex decision-making systems, we believe the benchmarks used in our study are well-suited to demonstrate the strengths and contributions of our method.
> > >
> > > Our primary focus is to enhance the decision-making capabilities of smaller open-source LLMs (e.g., 7B/8B models). The benchmarks we selected, such as ALFWorld, BabyAI-Text, and WebShop, provide controlled and widely-recognized environments for evaluating sequential reasoning and policy improvement. These settings allow us to clearly isolate and quantify the impact of our proposed method, as demonstrated by significant performance gains over strong baselines.
> > >
> > > We recognize that benchmarks like WebArena and OS World involve additional complexities, including diverse observation modalities and scaling trajectory rollouts. These benchmarks present significant challenges even for the most advanced closed-source models. For instance, GPT-4 achieves an end-to-end task success rate of only 11.70% in WebArena, compared to the human benchmark of 78.24% [2], and a success rate of 12.24% in OS World, compared to 72.36% for humans [3]. These results illustrate the difficulty of these tasks and underscore the gap that remains, even for leading models.
> > >
> > > While these benchmarks are undoubtedly valuable, we believe that the current benchmarks sufficiently validate the key contributions of our work. Future exploration of more complex settings like these benchmarks would require significantly more computational resources and model scaling, which were beyond the scope of this work. Nevertheless, the demonstrated improvements in decision-making on our chosen benchmarks highlight the broad potential of our method for diverse applications.

---

> > > ### Author Response · Authors · 2024-11-30
> > > **New Experimental Results and Clarifications to Other Questions (Part 4)**
> > >
> > > ### **Question 3:** The details of the computation analysis experiment.
> > >
> > > > **Question 3.1:** While the authors present additional results of computation analysis in Section 5.4, it seems to be a red flag to me as many important details are missing. E.g. What is the base model used here, are they the same for all methods?
> > >
> > > **A**: All results in Section 5.4 are based on the Llama3.1-8B model, and the same base model was used for all methods to ensure consistency.
> > >
> > > > **Question 3.2:** What is the hardware for this to be tested on, is there any necessary parallelism?
> > >
> > > **A**: All experiments were conducted on a single Nvidia A100 GPU with 80GB of video memory, supported by a machine with 100GB of RAM. To ensure a fair comparison across our experiments, we utilized the same version of the HuggingFace Transformers library [12], which employs consistent parallelism techniques for accelerating inference.
> > >
> > > > **Question 3.3:** How is each metric calculated?
> > >
> > > **A**: Each metric is calculated as follows:
> > >
> > > - `Steps per Task`:  This metric quantifies the number of actions executed for each task. It specifically counts the actions taken by our method and the baseline methods, excluding any planning or reasoning steps that may precede those actions.
> > > - `Time Used per Task`: This metric measures the total time taken to execute each task, from initiation to completion or termination. It encompasses all time costs associated with the task, including execution time as well as any time spent on reasoning and planning.
> > > - `Token Cost per Task`: This metric captures the total number of tokens utilized by each method during the execution of a task. It includes tokens consumed for both action generation and any planning activities that occur during the task execution.
> > >
> > > > **Question 3.4:** Why does ReAct have more steps but much less time per task?
> > >
> > > **A**: As we explained above, the `Steps per Task` metric counts only the execution steps involved in completing a task, while `Time Used per Task` metric encompasses the total time required, including thinking (including reasoning and planning) and execution phases.
> > >
> > > ReAct does not perform explicit planning, thus having sinificantly less thinking time. However, it can result in more execution steps as it may take longer to complete the task or may terminate when the maximum time step is reached. Thus, ReAct exhibits more steps but less overall time per task.
> > >
> > > > **Question 3.5:** Does "steps per task" take into account the costs of estimating Q(s,a) with trajectory rollouts?
> > >
> > > **A**: No. As mentioned above, `Steps per Task` counts how many actions are actually executed for each task. The planning or reasoning steps of our method and baselines are not taken into account for this metric.
> > >
> > > Thanks again for your prompt feedback. We hope our extra explanations and experimental results have made the novelty of our work more clear to you. We would appreciate it if you could re-evaluate our paper. If you have any further questions or concerns, please feel free to let us know.

---

> > > ### Author Response · Authors · 2024-11-30
> > > **New Experimental Results and Clarifications to Other Questions (Part 5)**
> > >
> > > **References**
> > >
> > > [1] Setlur, Amrith, et al. "RL on Incorrect Synthetic Data Scales the Efficiency of LLM Math Reasoning by Eight-Fold." arXiv preprint arXiv:2406.14532 (2024).
> > >
> > > [2] Zhou, Shuyan, et al. "WebArena: A Realistic Web Environment for Building Autonomous Agents." The Twelfth International Conference on Learning Representations.
> > >
> > > [3] Xie, Tianbao, et al. "Osworld: Benchmarking multimodal agents for open-ended tasks in real computer environments." arXiv preprint arXiv:2404.07972 (2024).
> > >
> > > [4] Mudgal, Sidharth, et al. "Controlled Decoding from Language Models." Forty-first International Conference on Machine Learning.
> > >
> > > [5] Rafailov, Rafael, et al. "Direct preference optimization: Your language model is secretly a reward model." Advances in Neural Information Processing Systems 36 (2024).
> > >
> > > [6] Verma, Mudit, Siddhant Bhambri, and Subbarao Kambhampati. "Do Think Tags Really Help LLMs Plan? A Critical Evaluation of ReAct-Style Prompting." Adaptive Foundation Models: Evolving AI for Personalized and Efficient Learning. NeurIPS Workshop (2024).
> > >
> > > [7] Shinn, Noah, et al. "Reflexion: Language agents with verbal reinforcement learning." Advances in Neural Information Processing Systems 36 (2024).
> > >
> > > [8] Carta, Thomas, et al. "Grounding large language models in interactive environments with online reinforcement learning." International Conference on Machine Learning. PMLR, 2023.
> > >
> > > [9] Chen, Wentse, et al. "Fine-tuning LLM Agents with Retrospective In-Context Online Learning." Adaptive Foundation Models: Evolving AI for Personalized and Efficient Learning. NeurIPS Workshop (2024).
> > >
> > > [10] Zhou, Andy, et al. "Language Agent Tree Search Unifies Reasoning, Acting, and Planning in Language Models." Forty-first International Conference on Machine Learning.
> > >
> > > [11] Yang, Hui, Sifu Yue, and Yunzhong He. "Auto-gpt for online decision making: Benchmarks and additional opinions." arXiv preprint arXiv:2306.02224 (2023).
> > >
> > > [12] Wolf, Thomas, et al. "Transformers: State-of-the-art natural language processing." Proceedings of the 2020 conference on empirical methods in natural language processing: system demonstrations. 2020.

---

> ### Author Response · Authors · 2024-11-30
> **New Experimental Results and Clarifications to Other Questions (Part 2)**
>
> > **Question 1.2:** For the value critic, it seems that the main difference from constrained decoding is that the value critic uses Monte Carlo estimations of the Q function instead of actually training a parametric Q function. This practice is also commonly adopted when it is unreliable to train a Q function (e.g. [1] https://arxiv.org/pdf/2406.14532).
>
> **A**: We also respectfully disagree with the claim about value-critic.
>
> (1) Our value-critic employs a unique Q-value estimation method described in Eq.3, which leverages the internal beliefs of LLMs about success and failure to compute the Q-value. This approach is distinct from the methodology presented in [1], which does not incorporate our specific design.
>
> (2) Though the KL-regularized RL objective is a common practice for balancing conflicting goals (as seen in methods like DPO [5] and Constrained Decoding [4]), the critical aspect lies in how each term in the objective is defined.  Our value-critic accounts for long-term consequences, resulting in more stable value-based evaluations derived from the LLMs' internal beliefs regarding success and failure.
>
> To further demonstrate the strengths of our value-critic. We conducted additional experiments comparing the performance of `LAC` and `LAC w/ direct evaluation` on the WebShop benchmark.
>
> - `LAC`: Our original method. It generates value-based evaluations by extracting LLMs' internal beliefs of success and failure as described in Eq.3.
> - `LAC w/ direct evaluation`: We prompt the LLMs to directly output the probability of success $p(y=+1)$ while keeping all other components unchanged. The Q-value is then calculated as $\log\frac{p(y=+1)}{1-p(y=+1)}$.
>
> The results, presented in Tables 7 and 8, show that our `LAC` method outperforms `LAC w/ direct evaluation` in terms of success rate and reward across most base models. Analysis of the resulting trajectories revealed that `LAC w/ direct evaluation` often produces a non-informative success probability (e.g., $p(y=+1)=0.5$), leading to ineffective improvements in policy.
>
>
> **Table 7: Success rate comparison of `LAC` and `LAC w/ direct evaluation` in WebShop benchmark**
>
> | Success Rate             | CodeLlama-7B | Gemma-7B | Llama-3-8B | Mistral-7B |
> | ------------------------ | ------------ | -------- | ---------- | ---------- |
> | LAC (ours)               | 32%          | **46%**  | **39%**    | **32%**    |
> | LAC w/ direct evaluation | **38%**      | 42%      | 22%        | 24%        |
>
> **Table 8: Final reward comparison of `LAC` and `LAC w/ direct evaluation` in WebShop benchmark**
>
> | Reward                   | CodeLlama-7B | Gemma-7B   | Llama-3-8B | Mistral-7B |
> | ------------------------ | ------------ | ---------- | ---------- | ---------- |
> | LAC (ours)               | **0.5840**   | **0.7237** | **0.6733** | 0.6299     |
> | LAC w/ direct evaluation | 0.5636       | 0.6975     | 0.6453     | **0.6333** |
>
>
>
> > **Question 1.3:** It is unclear to me what the challenges are to combine those two critics, and simply combining two common practices is not novel enough for a paper that claims to come up with a new method.
>
> **A**: We would like to clarify that our approach is not a combination of the critics. Instead, LAC represents a novel integration of the actor and dual-critics in a cohesive manner. The primary challenge we address is the seamless integration of the actor with both critics. Specifically, the novelty lies in:
>
> - LAC provides a **synergistic framework** where the critics are designed to complement each other—language critic captures contextual feedback from its history with interpretability and qualitative reasoning, while value critic evaluates predicted future trajectories and ensures quantitative alignment with long-term outcomes.
>
> - LAC's novel gradient-free optimization strategy harmonizes their outputs and seamlessly leverages both historical performance and future predictions to improve the actor's policy effectively.

---

### Official Review · Reviewer_thbr · 2024-11-04

**Soundness:** 3
**Presentation:** 3
**Contribution:** 3
**Rating:** 6
**Confidence:** 3

**Summary:**

The paper proposes a new way to tune LLMs to behave as agents, by combining the initial action probabilities with predictions by a language and value critic. The language critic adds natural language feedback to various candidate actions, and the value critic uses search to assign a value (or probability of success) to those actions. The method achieves impressive empirical performance on popular benchmarks such as ALFWorld against actor-only and critic-only baselines.

**Strengths:**

The paper is clearly written and tackles an important problem, as many applications of LLMs rely on them to behave as long-term agents rather than simply generate responses.

The method is sensible, using the language critic to refine the action space, then combining predictions by the value critic with the base LLM policy via a simple perturbation of initial action probabilities.

Finally, The empirical results are impressive, outperforming previous state-of-the-art techniques such as ReAct by a large margin.

**Weaknesses:**

Though the method is impressive, I have some concerns about its generalizability.

Namely, the authors only consider tasks with small action spaces, where evaluating each action individually is tractable. In many more realistic tasks such as dialogue, I imagine that the action space would be more open-ended and am unsure how to adapt the proposed method.

In addition, the tasks considered rely on being able to simulate future trajectories with high fidelity, which may be harder to do in more complex environments. Specifically, it is likely much harder to faithfully predict trajectories when the agent is engaging with another human rather than simply moving around in a static environment.

Finally, the value critic currently only works for tasks with binary outcomes (success or failure).

**Questions:**

Overall, I think the paper makes a valuable contribution. However, I do think there are several areas that the authors could address to further strengthen it:

(1) How would the method change when the task has a potentially infinite action space (such as in dialogue)?

(2) Have the authors experimented with tasks with more expressive outcomes/rewards? I am curious if both critic can still behave well when there are more nuanced outcomes than just success or failure.

(3) While the benchmarks are well-known, I think they are perhaps not realistic of tasks that people might actually want LLMs to accomplish. For example, it would be interesting to see the authors evaluate on tasks considered in the GDP-Zero paper such as donation solicitation [1].

[1] https://arxiv.org/abs/2305.13660

---

> ### Author Response · Authors · 2024-11-25
> **Additional Experimental Results and Clarifications to Other Questions (Part 1)**
>
> Thanks for the valuable comments and helpful suggestions. Here we provide additional experimental results and detailed explanations for your questions. The detailed experimental results are provided in the revised version of the paper.
>
> > **Weakness 1 & Question 1**: (W1:) Namely, the authors only consider tasks with small action spaces, where evaluating each action individually is tractable. In many more realistic tasks such as dialogue, I imagine that the action space would be more open-ended and I am unsure how to adapt the proposed method. (Q1:) How would the method change when the task has a potentially infinite action space (such as in dialogue)?
>
> **A**: Thanks for the insightful suggestion. We have conducted new experiments using the WebShop benchmark [2], which presents a scenario with a potentially infinite action space. This benchmark requires an agent to purchase a product based on specific instructions (e.g. "I need a long clip-in hair extension which is natural looking, and price lower than 20.00 dollars") through web interactions (e.g. search "long clip-in hair extension", click buttons such as "[item ID]" or "back to search"). Within this context, the 'search' and 'click' actions can indeed lead to an unbounded set of potential actions, as the agent can continuously refine its queries and selections based on dynamic web results.
>
> We represent the detailed results in Figure 7 of the revised paper, and we also show some results in Table 1 and Table 2 below.  We found that our method, LAC, consistently outperforms other baselines, in terms of both success rate and final reward across various base models. This demonstrates the robustness of our method in handling more complex and open-ended action spaces.
>
> **Table 1: Success rate comparison in WebShop benchmark**
> | Success Rate | CodeLlama-7B | Gemma-7B | Llama-3-8B | Mistral-7B |
> | ------------ | ------------ | -------- | ---------- | ---------- |
> | LAC (ours)   | **32%**      | **46%**  | **39%**    | **32%**    |
> | ReAct        | 15%          | 35%      | 37%        | 30%        |
> | RAP          | 19%          | 28%      | 28%        | 26%        |
>
> **Table 2: Final reward comparison in WebShop benchmark**
> | Reward     | CodeLlama-7B | Gemma-7B   | Llama-3-8B | Mistral-7B |
> | ---------- | ------------ | ---------- | ---------- | ---------- |
> | LAC (ours) | **0.5840**   | **0.7237** | **0.6733** | **0.6299** |
> | ReAct      | 0.5042       | 0.6332     | 0.6445     | 0.6159     |
> | RAP        | 0.5545       | 0.6048     | 0.6215     | 0.5594     |
>
> > **Weakness 2**:  In addition, the tasks considered rely on being able to simulate future trajectories with high fidelity, which may be harder to do in more complex environments. Specifically, it is likely much harder to faithfully predict trajectories when the agent is engaging with another human rather than simply moving around in a static environment.
>
> **A**: We acknowledge the challenge of accurately predicting future trajectories in complex environments. To mitigate this issue, our method incorporates a KL-divergence constraint in Equation 4, balancing the actor's prior knowledge with the critic's evaluations, rather than solely relying on the critic's evaluations. This approach reduces reliance on potentially inaccurate future predictions.
>
> > **Weakness 3 & Question 2**:  (W1:) Finally, the value critic currently only works for tasks with binary outcomes (success or failure). (Q2:) Have the authors experimented with tasks with more expressive outcomes/rewards? I am curious if both critics can still behave well when there are more nuanced outcomes than just success or failure.
>
> **A**: Thank you for your valuable feedback. The WebShop benchmark discussed above also includes expressive rewards in the range of [0, 1]. In scenarios where the purchased product only partially satisfies requirements, the final reward is a value between 0 and 1. The results of the final reward in Table 2 above show that LAC outperforms other baselines across various base models, demonstrating that it can effectively adapt to more nuanced outcomes beyond binary success or failure.

---

> ### Author Response · Authors · 2024-11-25
> **Additional Experimental Results and Clarifications to Other Questions (Part 2)**
>
> > **Question 3**:  While the benchmarks are well-known, I think they are perhaps not realistic of tasks that people might actually want LLMs to accomplish. For example, it would be interesting to see the authors evaluate on tasks considered in the GDP-Zero paper such as donation solicitation [1].
>
> **A**: Thank you for suggesting an evaluation on more realistic tasks. While we were unable to include results on donation solicitation as discussed in [3] due to time constraints, we did conduct experiments on WebShop [2], which we believe represents a more realistic task that LLMs might be expected to perform. A brief description of this benchmark and the corresponding results are provided in the responses above. Our proposed method, LAC, consistently outperforms other baselines in terms of both success rate and final reward across various base models. These results highlight the effectiveness of our approach in addressing more complex and realistic tasks. We hope the experiments on WebShop [2] address your concerns.
>
> We hope these revisions and clarifications adequately address your concerns. Thank you for your valuable feedback and insightful suggestions.
>
> ---
>
> **References**
>
> [1] Yu, Xiao, Maximillian Chen, and Zhou Yu. "Prompt-Based Monte-Carlo Tree Search for Goal-oriented Dialogue Policy Planning." The 2023 Conference on Empirical Methods in Natural Language Processing.
>
> [2] Yao, Shunyu, et al. "Webshop: Towards scalable real-world web interaction with grounded language agents." Advances in Neural Information Processing Systems 35 (2022): 20744-20757.
>
> [3] Yu, Xiao, Maximillian Chen, and Zhou Yu. "Prompt-Based Monte-Carlo Tree Search for Goal-oriented Dialogue Policy Planning." arXiv preprint arXiv:2305.13660 (2023).

---

> ### Author Response · Authors · 2024-11-26
> **Follow-Up on Rebuttal for Your Review**
>
> Dear Reviewer,
>
> Thank you for your time and effort in reviewing our work. We have provided detailed clarifications and experimental results in our rebuttal to address the issues and concerns raised in your comments.
>
> If our response satisfactorily resolves your concerns, we kindly ask if you could reconsider your evaluation of our work. Should you have any additional questions or comments, we would be happy to engage in further discussions to ensure all aspects are addressed.
>
> Thank you again for your thoughtful review and support.
>
>
> Best regards,
> The Authors

---

> ### Comment · Reviewer_thbr · 2024-11-27
> **Reviewer thbr Response**
>
> Thank you for conducting additional experiments on the WebShop domain. Could the authors clarify how actions were chosen to be "judged" when the action space is large? Were a subset of permissible actions sampled at the beginning?
>
> Overall, I still believe that the paper makes a positive contribution and maintain my score. I do find the results between LAC and ReAct to be rather close for more sophisticated models though. They also appear to be weaker than the results obtained by LATS, which to my knowledge, currently achieves state-of-the-art in this domain [1]. If the authors could also show a substantial benefit of LAC over LATS, I would consider further raising my score.
>
> [1] https://arxiv.org/pdf/2310.04406

---

> ### Author Response · Authors · 2024-11-29
> **New Experimental Results and Clarifications to Other Questions**
>
> Thank you for your timely feedback! Here we provide new experimental results and extra clarifications to your questions.
>
> > **Question 1:** Could the authors clarify how actions were chosen to be "judged" when the action space is large? Were a subset of permissible actions sampled at the beginning?
>
> **A**: We do not manually sample a subset of permissible actions at the beginning. Instead, we allow the actor to sample a subset of candidate actions with higher probabilities and then apply gradient-free policy improvement to refine the action distribution. This strategy enables us to focus on actions with the highest potential, allowing us to efficiently navigate the large action space while ensuring effective decision-making.
>
>
> > **Question 2:** I do find the results between LAC and ReAct to be rather close for more sophisticated models though.
>
> **A**:  We appreciate your observation. However, it's important to note that the performance of LAC and ReAct is influenced by a variety of factors of LLMs beyond model complexity, including reasoning, modeling, and planning capabilities. For instance, as shown in the results for WebShop and AlfWorld, the `final reward` or `success rate` gap between LAC+Llama-3-8B and ReAct+Llama-3-8B is more significant than the gap between LAC+Mistral-7B and ReAct+Mistral-7B, even though Llama-3-8B is more sophisticated than Mistral-7B. This suggests that the disparity in performance is not solely a function of the sophistication of the models used.
>
>
> > **Question 3:** They also appear to be weaker than the results obtained by LATS, which to my knowledge, currently achieves state-of-the-art in this domain [1]. If the authors could also show a substantial benefit of LAC over LATS, I would consider further raising my score.
>
> **A**: Thanks for your constructive suggestion. We present a detailed performance comparison between LAC and LATS in Table 3 and Table 4 below. Our results demonstrate that LAC consistently outperforms LATS across all evaluated base models on the WebShop benchmark. Notably, LAC combined with Gemma-7B surpasses the performance of LATS with GPT-3.5 in terms of Success Rate and achieves comparable performance in terms of Reward. These findings underscore the effectiveness of LAC in enhancing the decision-making capabilities of large language models.
>
>
> **Table 3: Success rate comparison between LAC and LATS in WebShop**
>
> | Success Rate                     | CodeLlama-7B | Gemma-7B | Llama-3-8B | Mistral-7B |
> | :------------------------------- | :----------- | :------- | :--------- | :--------- |
> | LAC (ours)                       | **32%**      | **46%**  | **39%**    | **32%**    |
> | LATS (reported: 38%, w/ GPT-3.5) | 14%          | 39%      | 37%        | 27%        |
> | ReAct                            | 15%          | 35%      | 37%        | 30%        |
> | RAP                              | 19%          | 28%      | 28%        | 26%        |
>
>
> **Table 4: Final reward comparison between LAC and LATS in WebShop**
>
> | Reward                            | CodeLlama-7B | Gemma-7B   | Llama-3-8B | Mistral-7B |
> | :-------------------------------- | :----------- | :--------- | :--------- | :--------- |
> | LAC (ours)                        | **0.5840**   | **0.7237** | **0.6733** | **0.6299** |
> | LATS (reported: 75.9, w/ GPT-3.5) | 0.4924       | 0.6313     | 0.6521     | 0.6063     |
> | ReAct                             | 0.5042       | 0.6332     | 0.6445     | 0.6159     |
> | RAP                               | 0.5545       | 0.6048     | 0.6215     | 0.5594     |
>
> Thanks again for your prompt feedback. We hope our extra explanations and experimental results address your concerns and we would appreciate it if you could re-evaluate our paper. If you have any further questions or concerns, please feel free to let us know.
>
>
> ---
>
> **References**
>
> [1] Zhou, Andy, et al. "Language Agent Tree Search Unifies Reasoning, Acting, and Planning in Language Models." Forty-first International Conference on Machine Learning.

---

### Official Review · Reviewer_FHde · 2024-11-04

**Soundness:** 3
**Presentation:** 2
**Contribution:** 2
**Rating:** 5
**Confidence:** 4

**Summary:**

Although large language models (LLMs) have shown impressive capabilities in natural language processing tasks, they struggle with complex reasoning tasks. One common approach to tackle this issue is to train LLMs using reinforcement learning (RL); however, RL methods have several drawbacks. The authors propose a novel gradient-free Actor-Critic framework based on LLMs to overcome these limitations. This new framework includes an actor and a critic component, distinguishing it from previous approaches. Notably, the Critic is designed to provide feedback in either language or numerical form to help update the actor.

**Strengths:**

- The ability to help LLMs reason in complex decision-making scenarios is a very important task.
- The structure of the paper was well organized and easy to follow, aside from some terminology framing issues.
- The paper effectively demonstrates how different types of actor feedback—reasoning, evaluation, and future trajectory prediction—affect downstream performance.

**Weaknesses:**

- The terminology used in the paper could be more precise, particularly in relation to terms commonly found in the reinforcement learning literature.
- The paper is missing important baselines needed to understand the performance gain claims.

**Questions:**

- Is it reasonable to summarize the algorithm differences in the following manner: ReAct includes reasoning + actor, Critic only includes future trajectory + actor, and Lang-critic includes evaluation + actor?
- Why does the value critic require a future trajectory, and how does it perform without future trajectories?
- How does ReAct, combined with a value critic, perform?
- How does ReAct, combined with a language critic, perform?
- Is the prior \pi_{LLM} the same LLMs used for Q_{LLM} for computing the critic values?
- How does language critic + value critic perform? (Essentially LAC without update action distribution, instead using the critic values to choose an action)

---

> ### Author Response · Authors · 2024-11-25
> **New Experimental Results and Clarifications of the Questions (Part 1)**
>
> Thanks for your comments and valuable suggestions. Here we provide detailed explanations for your questions. The detailed experimental results are provided in the revised version of our paper.
>
> > **Question 1**: Is it reasonable to summarize the algorithm differences in the following manner: ReAct includes reasoning + actor, Critic only includes future trajectory + actor, and Lang-critic includes evaluation + actor?
>
> **A**: For better clarification, we first provide a summary of our method’s variants in Table 1 below. Our approach integrates the actor with dual-critics: the Lang-Critic, which provides language-based evaluations of actions, and the Value-Critic, which offers value-based assessments of candidate actions that will be used to refine the policy via gradient-free policy improvement as described in Eq.6.
>
> Regarding the differences:
>
>  - ReAct includes reasoning + actor, but it does not include critics so we classify it into `Actor-only` methods.
>  - The term `Critic-only` is misleading if interpreted as including future trajectory + actor. `Critic-only` refers to the methods that solely rely on value-based evaluations of actions for decision-making while ignoring the actor's prior knowledge of action distribution, although they also use the `actor` to sample candidate actions.
>  - Similarly, characterizing `Lang-critic` as including evaluation + actor is inaccurate. The `Lang-Critic` is a component of our LAC that specifically provides language-based evaluations of actions.
>
> **Table 1: Summary of our method's variants.**
> | Method               | Actor   | Lang-Critic | Value-Critic                                  |
> | -------------------- | ------- | ----------- | --------------------------------------------- |
> | LAC                  | $\surd$ | $\surd$     | $\surd$                                       |
> | LAC w/o rollout      | $\surd$ | $\surd$     | $\surd\times$ (w/o rollout)                   |
> | Value-critic only    | $\surd$ | $\surd$     | $\surd\times$ (w/o policy improvement, Eq.6)  |
> | LAC w/o value-critic | $\surd$ | $\surd$     | $\times$                                      |
> | LAC w/o lang-critic  | $\surd$ | $\times$    | $\surd$                                       |
> | ReAct                | $\surd$ | $\times$    | $\times$                                      |
>
> > **Weakness 2 & Question 2**: (W2:) The paper is missing important baselines needed to understand the performance gain claims. (Q2:) Why does the value critic require a future trajectory, and how does it perform without future trajectories?
>
> **A**: The `Value-Critic` relies on future trajectory predictions to improve the accuracy of its evaluations. By predicting future trajectories, the critic considers long-term consequences and evaluates actions more effectively, which ultimately leads to better decision-making.
>
> For a full comparison, we have conducted a new experiment for `LAC w/o rollout`, in which the `Value-Critic` generate value-based evaluations without future trajectory predictions. We represent the detailed result in Figure 10 and Figure 11 of the revised paper, and we also show some results in Table 2 and Table 3 below. The results show that `LAC w/o rollout` consistently underperforms compared to the full LAC across various base models. This finding emphasizes the importance of future trajectory predictions for accurate evaluations.
>
> **Table 2: More ablation studies in AlfWorld benchmark.**
> | Method               | CodeLlama-7B | Gemma-7B   | Llama-3-8B | Mistral-7B |
> | -------------------- | ------------ | ---------- | ---------- | ---------- |
> | LAC                  | **79.10%**   | **84.33%** | **77.61%** | **79.10%** |
> | LAC w/o rollout      | 59.70%       | 74.63%     | 31.34%     | 64.67%     |
> | Value-critic only    | 47.01%       | 64.69%     | 66.42%     | 66.42%     |
> | LAC w/o value-critic | 71.64%       | 72.39%     | 58.21%     | 76.12%     |
> | LAC w/o lang-critic  | 41.79%       | 70.15%     | 47.76%     | 56.72%     |
> | ReAct                | 20.15%       | 54.48%     | 30.60%     | 33.83%     |
>
> **Table 3: More ablation studies in BabyAI benchmark.**
> | Method               | CodeLlama-7B | Gemma-7B | Llama-3-8B | Mistral-7B |
> | -------------------- | ------------ | -------- | ---------- | ---------- |
> | LAC                  | **46%**      | **76%**  | **66%**    | **70%**    |
> | LAC w/o rollout      | 44%          | 60%      | 64%        | 50%        |
> | Value-critic only    | 38%          | 64%      | 56%        | 62%        |
> | LAC w/o value-critic | 34%          | 52%      | 52%        | 38%        |
> | LAC w/o lang-critic  | 32%          | 60%      | 48%        | 54%        |
> | ReAct                | 42%          | 48%      | 42%        | 26%        |

---

> ### Author Response · Authors · 2024-11-25
> **New Experimental Results and Clarifications of the Questions (Part 2)**
>
> > **Question 3**: How does ReAct, combined with a value critic, perform?
>
> **A**: As indicated in Table 1 above, ReAct combined with a Value-Critic corresponds to `LAC w/o lang-critic`. The performance results for this configuration are presented in Figure 4 of the initial submission (Section 5.3). In brief, `LAC w/o lang-critic` consistently underperforms compared to the full LAC in both AlfWorld and BabyAI benchmarks across various base models. However, it generally outperforms ReAct, highlighting the effectiveness of our value-critic.
>
> > **Question 4**: How does ReAct, combined with a language critic, perform?
>
> **A**: Similarly, ReAct combined with a Lang-Critic corresponds to `LAC w/o value-critic`. The results for this configuration are also shown in Figure 4 of the initial submission (Section 5.3). In summary, `LAC w/o value-critic` underperforms compared to the full LAC across various base models in the AlfWorld and BabyAI benchmarks. Nonetheless, it typically outperforms ReAct alone, indicating the effectiveness of our lang-critic.
>
> > **Question 5**: Is the prior \pi_{LLM} the same LLMs used for Q_{LLM} for computing the critic values?
>
> **A**: No, they are not identical. $\pi_{LLM}$ is based on the original model, while $\mathcal{Q}_{LLM}$ utilizes a fine-tuned version of the same model, adapted using several trajectories. The fine-tuning details are provided in Appendix B.2.
>
> > **Question 6**: How does language critic + value critic perform? (Essentially LAC without update action distribution, instead using the critic values to choose an action)
>
> **A**: According to Table 1 above, the configuration of the language critic combined with the value critic corresponds to `Value-critic only`. The performance results for this setup are detailed in Figure 4 of the initial submission. In summary, `Value-critic only` consistently underperforms full LAC, demonstrating the effectiveness of our gradient-free policy improvement component.
>
> > **Weakness 1**:  The terminology used in the paper could be more precise, particularly in relation to terms commonly found in the reinforcement learning literature.
>
> **A**: Thank you for your feedback. If the reviewer could specify which terms you find imprecise, it would greatly help us improve the clarity of our paper.
>
> Thanks again for your efforts and insightful comments! We hope our clarification addresses your concerns and sincerely appreciate it if you could re-evaluate our work. Any further feedback and discussions are much appreciated.

---

> > ### Comment · Reviewer_FHde · 2024-11-26
> >
> > I acknowledge the author's response and thank them for addressing the weaknesses and questions I raised. I will maintain my score.

---

### Official Review · Reviewer_KnwK · 2024-11-09

**Soundness:** 2
**Presentation:** 2
**Contribution:** 3
**Rating:** 6
**Confidence:** 4

**Summary:**

Authors propose a self-reflecting flow architecture for multi-step problem solving which is informed by classical RL concepts.
On AlfWorld and BabyAI-text environments it exhibits good performance.

Although components of the architecture have been previously proposed, the combination is sensible.
* "Lang-critic": this components reflects upon the goal and the history and augments the prompt for the actor component.
* "actor": proposes new actions given the goal, the history, and the lang-critic augmentation.
* "rollout simulator": given a goal, history, and action: simulates the next few steps (under what policy?)
* "value-critic": given a goal, history, proposed action, and simulated future under this action: estimate the likelihood of task completion

Multiple actions are sampled from the actor at each round, scored by the value critic, the distribution is reweighted via exponential-log-odds, and then the greedy action is selected.

There are some ablations studies to provide insight into the importance of the components, and comparisons to classical RL techniques.

**Strengths:**

In a zero-shot context, the architecture and results are facially reasonable, given there are numerous prior results indicating large foundation models can exhibit improved performance when composed via a self-reflecting architecture, and given prior art that reasonable synthetic rollouts can improve value estimation analogous to chain-of-thought for possible futures.

On balance, despite the poor exposition around fine-tuning, I believe readers would net benefit from exposure to this paper because of intriguing concepts such as: 1) exponential log-odds reweighting of a prior action distribution is superior to policy gradient for sculpting the action distribution in the small-sample regime [line 375 vs. line 833]; 2) foundational [rather than component or architecture specific] fine-tuning induces end-to-end improvement when composed [lines 1058-1066].

**Weaknesses:**

This paper has two kinds of weaknesses.

The first kind is due to the nature of academic work, which is resource constrained (small teams; limited compute).   This induces a set of "easy" criticisms such as "insufficient experimental validation" or "excessive focus on the small sample regime".  I believe both authors and readers are well-aware of practical constraints, so this reviewer will not weigh such concerns heavily.

The second kind is insufficient description to allow the reader to understand what was done.  Specifically, the weakest parts of the paper are all related to the impact of fine-tuning, which is not sufficiently described (see questions).  Authors could improve both the intelligibility and the impact of this paper via more detail.

**Questions:**

1. Authors mention fine-tuning under a limited budget (e.g., 18 trajectories), following the specification on lines 1048-1057.  However what is the source these trajectories?  Are they human annotated trajectories?  Trajectories sampled from the zero-shot architecture, perhaps conditioned on task success?  How are the special tokens which indicate positive/negative judgement produced for the fine-tuning data?  It is not clear, and in the small-sample regime, these details are critical.
   1. Related: Under what policy (action distribution) is the future simulator generating?  If the reweighted action distribution becomes divergent from the prior, will the future simulator be invalidated?
1. Authors argue equation (6) on line 375 is a more sample efficient way to incorporate labeled trajectory information than conventional policy gradient via the equation on line 833.  However the value-critic on line 833 does not take simulated rollouts as a parameter.  Perhaps this is a typo?  Otherwise the comparison is invalid.
1. In figure 8, it is unclear why LAC+fine-tuned-actor underperforms LAC.  The lack of commentary raises reader suspicion.

---

> ### Author Response · Authors · 2024-11-25
> **Clarifications of the Questions (Part 1)**
>
> Thanks for the valuable comments and helpful suggestions. Here we provide detailed explanations for your questions.
>
> > **Weakness 1**:  The first kind is due to the nature of academic work, which is resource constrained (small teams; limited compute). This induces a set of "easy" criticisms such as "insufficient experimental validation" or "excessive focus on the small sample regime".
>
> **A**: Thank you for your understanding regarding the constraints of academic work. While we acknowledge the limitations of resources, our study demonstrates that even with few samples and small models, it is possible to achieve performance that outperforms state-of-the-art LLMs.
>
> To further validate the generalizability of our method, we have conducted new experiments using the WebShop benchmark [1], which simulates a web browsing task with more complex and infinite action spaces. In this scenario, an agent is required to purchase a product based on specific instructions (e.g. "I need a long clip-in hair extension which is natural looking, and price lower than 20.00 dollars") through web interactions (e.g. search "long clip-in hair extension", click buttons such as "[item ID]" or "back to search"). Within this context, the 'search' and 'click' actions can indeed lead to an unbounded set of potential actions, as the agent can continuously refine its queries and selections based on dynamic web results.
>
> We represent the detailed results in Figure 7 of the revised paper, and we also show some results in the following Table 1 and Table 2. We found that our method, LAC, consistently outperforms other baselines, in terms of both success rate and final reward across various base models. This demonstrates the generalizability of our method in handling more complex and infinite action spaces.
>
> **Table 1: Success rate comparison in WebShop benchmark**
> | Success Rate | CodeLlama-7B | Gemma-7B | Llama-3-8B | Mistral-7B |
> | ------------ | ------------ | -------- | ---------- | ---------- |
> | LAC (ours)   | **32%**      | **46%**  | **39%**    | **32%**    |
> | ReAct        | 15%          | 35%      | 37%        | 30%        |
> | RAP          | 19%          | 28%      | 28%        | 26%        |
>
> **Table 2: Final reward comparison in WebShop benchmark**
> | Reward     | CodeLlama-7B | Gemma-7B   | Llama-3-8B | Mistral-7B |
> | ---------- | ------------ | ---------- | ---------- | ---------- |
> | LAC (ours) | **0.5840**   | **0.7237** | **0.6733** | **0.6299** |
> | ReAct      | 0.5042       | 0.6332     | 0.6445     | 0.6159     |
> | RAP        | 0.5545       | 0.6048     | 0.6215     | 0.5594     |
>
> > **Question 1**:  Authors mention fine-tuning under a limited budget (e.g., 18 trajectories), following the specification on lines 1048-1057. However what is the source these trajectories? Are they human annotated trajectories? Trajectories sampled from the zero-shot architecture, perhaps conditioned on task success? How are the special tokens which indicate positive/negative judgment produced for the fine-tuning data?
>
> **A**: The 18 trajectories used for fine-tuning are human-annotated and sourced from the training split of our benchmarks, including both successful and failed trajectories to ensure balance. Each can be annotated in about 10-20 minutes, allowing us to maintain a manageable budget.
>
> The special tokens indicating positive and negative judgments for the fine-tuning data are also human-annotated. Specifically, we assigned the following labels based on the relevance of actions to the final goal:
>
> - Positive judgment like "GOOD": assigned to actions deemed necessary for achieving the final goal.
> - Negative judgment like "BAD": assigned to actions that were determined to be useless or incorrect in the context of reaching the goal.
> - Other judgments like "UNKNOWN": assigned to actions that could not be evaluated as either good or bad based on the trajectory history.
>
> We have included this information in Appendix C.3 of our paper for further reference, where we detail the annotation process and the criteria used for assigning these labels.
>
> > **Question 1.1**: Under what policy (action distribution) is the future simulator generating? If the reweighted action distribution becomes divergent from the prior, will the future simulator be invalidated?
>
> **A**: The future trajectories are generated based on the original actor $\pi_{LLM}$. While this may lead to some divergence of the reweighted action distribution from the prior, our method mitigates this risk by incorporating a KL-divergence constraint in Equation 4 in our manuscript, which helps to prevent the new actor from deviating too far from the original actor.
>
> ---
>
> **References**
>
> [1] Yao, Shunyu, et al. "Webshop: Towards scalable real-world web interaction with grounded language agents." Advances in Neural Information Processing Systems 35 (2022): 20744-20757.

---

> > ### Comment · Reviewer_KnwK · 2024-11-26
> > **I have read your response**
> >
> > WebShop results appear net positive.  I maintain my score but find the revisions helpful (e.g., typo fix mentioned in another response).
> >
> > > A: The future trajectories are generated based on the original actor
> > . While this may lead to some divergence of the reweighted action distribution from the prior, our method mitigates this risk by incorporating a KL-divergence constraint in Equation 4 in our manuscript, which helps to prevent the new actor from deviating too far from the original actor.
> >
> > Does this limit the amount of policy improvement that is possible?

---

> ### Author Response · Authors · 2024-11-25
> **Clarifications of the Questions (Part 2)**
>
> > **Question 2**:  Authors argue equation (6) on line 375 is a more sample efficient way to incorporate labeled trajectory information than conventional policy gradient via the equation on line 833. However the value-critic on line 833 does not take simulated rollouts as a parameter. Perhaps this is a typo? Otherwise the comparison is invalid.
>
> **A**: We thank the reviewer for pointing this out. The value-critic on line 833 also takes the simulated rollouts as input in practice. We have corrected this typo in the revised version of our paper.
>
> > **Question 3**:  In figure 8, it is unclear why LAC+fine-tuned-actor underperforms LAC. The lack of commentary raises reader suspicion.
>
> **A**: Compared to LAC, the underperformance of `LAC w/ fine-tuned-actor` arises from its tendency to overfit the training trajectories. This overfitting causes the actor to favor actions that are more frequent in the dataset, potentially leading to suboptimal action selection.
>
> For example, in the AlfWorld training dataset, the action "take an apple from X" occurs frequently. After fine-tuning, the actor may disproportionately generate this action, even when it is irrelevant to the current goal. One case is that the current goal is to "heat some egg and put it in the garbage can". When the agent sees an "apple 2" in "fridge 1", it generates and selects an irrelevant action "take apple 2 from fridge 1", which does not align with the task.
>
> This tendency towards overfitting arises because the complexity of the policy function, which maps states $s$ to actions $a$, often exceeds that of the critic. The policy often has to capture a wide variety of potential actions for each state, particularly in complex environments. However, the quite limited training dataset in our setting restricts its ability to generalize effectively, resulting in memorization of specific actions rather than flexible decision-making. In contrast, our critic, which includes a world model for rollout and an evaluation function, focuses on capturing more predictable dynamics of the environment and simpler evaluation criteria. This typically requires simpler mappings than those needed for the policy, thus avoiding overfitting.
>
> We have added this clarification in the revised version of our paper to address any ambiguity.
>
> We hope these revisions and clarifications adequately address your concerns. Thank you for your valuable feedback.

---

### Meta-Review · Area_Chair_Q5mg · 2024-12-21

**Metareview:**

This paper presents an approach to improve decision-making in LLMs. Given a current state and goal, the approach first generates language feedback on previous actions which is then used to .

The main concerns with this paper are: (i) lack of novelty and (ii) experiments being presented over older less-challenging methods than more recent challenging benchmarks like Web Arena. The novelty argument comes down to the fact that multiple papers now use the different pieces present in the description approach. There is a large body of work that uses LLMs to generate actions for decision-making, then there are LLM reflection papers that generate language feedback using the LLM and use them to improve the model, and finally, there are papers that generate reward values using LLM. This concern doesn't bother me too much as it can be non-trivial to assemble these ideas into a product. Implementation matters a lot, as I think the ML community has learned in the last few years. However, I agree with reviewer Xk7a that one would have expected results on more challenging domains. I think WebShop is a good domain that the authors have added during the discussion period, but results on challenging domains like WebArena would have helped a lot.

I do like the results here and the authors have done a good job overall in presenting more baselines in the discussion. The novelty part doesn't bother me too much. That said, I am leaning towards a weak rejection for now with a strong encouragement to submit again with results on more challenging domains. I think the paper is close to an accepted state.

**Additional Comments On Reviewer Discussion:**

Reviewers discussed this paper thoroughly. The main concerns that came up were:

1. Reviewer KnwK and Xk7a raised concerns on weak experiments: particularly lack of challenging domains. Authors added experiments on a new domain called WebShop and got positive results. In response, reviewer Xk7a argued that WebShop is somewhat contrived and experiments on OSSWorld or WebArena will be better. While I appreciate that authors added a new domain, I do think including a more common and challenging domain like WebArena would have been nice.

2. Comparisons with other approaches such as LATS (reviewer thbr) and variations of the algorithms (reviewer FHde). Authors present positive results against LATS and variations. Overall, I think this makes the approach promising.

3. There were concerns about novelty as past works have studied the different pieces in this paper. I am okay with the approach not being novel but this makes achieving positive results on challenging domains more necessary.

Overall, I think authors addressed concerns in (2) and I think lack of novelty in itself isn't that important in my assessment. But I think further evaluations are necessary to ensure that the approach works generally.

---

### Decision · Program_Chairs · 2025-01-22

Reject